# Short-term plasticity at cerebellar granule cell to molecular layer interneuron synapses expands information processing

Kevin Dorgans[1†], Valérie Demais[2], Yannick Bailly[1,2], Bernard Poulain[1], Philippe Isope[1], Frédéric Doussau[1]*

[1]Institut des Neurosciences Cellulaires et Intégratives, CNRS UPR 3212, Université de Strasbourg, Strasbourg, France; [2]Plateforme Imagerie in vitro, CNRS UPS 3156, Strasbourg, France

**Abstract** Information processing by cerebellar molecular layer interneurons (MLIs) plays a crucial role in motor behavior. MLI recruitment is tightly controlled by the profile of short-term plasticity (STP) at granule cell (GC)-MLI synapses. While GCs are the most numerous neurons in the brain, STP diversity at GC-MLI synapses is poorly documented. Here, we studied how single MLIs are recruited by their distinct GC inputs during burst firing. Using slice recordings at individual GC-MLI synapses of mice, we revealed four classes of connections segregated by their STP profile. Each class differentially drives MLI recruitment. We show that GC synaptic diversity is underlain by heterogeneous expression of synapsin II, a key actor of STP and that GC terminals devoid of synapsin II are associated with slow MLI recruitment. Our study reveals that molecular, structural and functional diversity across GC terminals provides a mechanism to expand the coding range of MLIs.

DOI: https://doi.org/10.7554/eLife.41586.001

**\*For correspondence:**
doussau@inci-cnrs.unistra.fr

**Present address:** †Okinawa Institute of Science and Technology Graduate University, Okinawa, Japan

**Competing interests:** The authors declare that no competing interests exist.

## Introduction

Inhibitory interneurons mediating feed-back or feed-forward inhibition (FFI) provide brain microcircuits with an exquisite temporal control over the firing frequency of projecting neurons (*Hennequin et al., 2017*; *Isaacson and Scanziani, 2011*; *Klausberger and Somogyi, 2008*; *O'Donnell et al., 1993*). In the cerebellar cortex, the FFI microcircuit is activated by granule cells (GCs) that target two types of molecular layer interneurons (MLIs): stellate cells (SCs) and basket cells (BCs). SCs and BCs finally control Purkinje cells (PC), the sole projecting neurons of the cerebellar cortex, through a powerful somatic or dendritic inhibition (*Jörntell et al., 2010*). In combination with the direct excitatory pathway provided by GC-PC connections, the FFI encodes sensorimotor information through acceleration or deceleration of PC simple spike activity (*Armstrong and Edgley, 1988*; *Jelitai et al., 2016*; *Ozden et al., 2012*).

Sensorimotor information is conveyed to the cerebellar cortex by mossy fibers (MFs) as short high-frequency bursts of action potentials (*Chadderton et al., 2004*; *Chen et al., 2017*; *Jörntell and Ekerot, 2006*; *Kennedy et al., 2014*; *Powell et al., 2015*; *Rancz et al., 2007*). During high-frequency stimulations, cerebellar synapses exhibit several forms of short-term synaptic plasticity (STP) including facilitation and depression of synaptic responses in the millisecond range (*Atluri and Regehr, 1996*; *Bao et al., 2010*; *Brachtendorf et al., 2015*; *Dittman et al., 2000*; *Doussau et al., 2017*; *Miki et al., 2016*; *Valera et al., 2012*; *Zheng and Raman, 2010*). STP play multiple roles in information processing (*Anwar et al., 2017*; *Fioravante and Regehr, 2011*). At the input stage of the cerebellar cortex, a strategy based on the heterogeneity of STP across MF-GC synapses provides a mechanism for coding multisensory events at the level of single GCs

(*Chabrol et al., 2015*). Also, differences in the profile of STP across synapses involved in the direct excitatory pathway or in the FFI microcircuit control the inhibitory/excitatory balance and shape Purkinje cell discharge (*Grangeray-Vilmint et al., 2018*). GCs which are the most numerous neurons in the brain, segregate in clonally related subpopulations (*Espinosa and Luo, 2008*). Despite such number and differences, STP heterogeneity across GC boutons is poorly documented. A seminal study has shown that the behavior of glutamate release during high-frequency activities at GC boutons is determined by the target cell (that is PC, SC or BC, *Bao et al., 2010*). Following compound stimulations of clusters of GCs or beams of parallel fibers (PFs), it was shown that GC-BC synapses depress during high-frequency stimulation while GC-SC synapses facilitate (*Bao et al., 2010*). By controlling the spatiotemporal excitability of PC (*Bao et al., 2010*), and potentially by shaping the inhibitory/excitatory balance (*Grangeray-Vilmint et al., 2018*), target cell–dependency of STP at the input stage of the FFI pathway must have important functional consequences for cerebellar output. However, target cell–dependency of STP at GC-MLI synapses is challenged by different experimental findings. First, many MLIs cannot be classified solely by their axon profile (e.g. basket versus dendritic synapses) or their position in the molecular layer (*Palay and Chan-Palay, 1974*; *Sultan and Bower, 1998*) and it was proposed that MLIs represent a single population of interneurons (*Jörntell et al., 2010*; *Rakic, 1972*; *Sotelo, 2015*; *Sultan and Bower, 1998*). Second, release properties and STP profiles of GC synaptic inputs to MLIs can be modified by presynaptic long-term plasticity and by local retrograde signaling independently of the target cell (*Bender et al., 2009*; *Soler-Llavina and Sabatini, 2006*).

Given the abundance of GCs and the importance of STP at GC-MLI synapses for cerebellar computation, we set out to study the diversity of STP at unitary GC-MLI synapses. Also, we aimed to uncover the molecular determinants of functional heterogeneity. Among presynaptic proteins involved in STP, synapsins (Syn) are good candidates underlying functional heterogeneity across cerebellar synapses. Syn are presynaptic phosphoproteins coded by three distinct genes (Syn I, II and III). Both Syn I and Syn II regulate neurotransmitter release and STP in mature synapses (*Cesca et al., 2010*; *Humeau et al., 2011*; *Song and Augustine, 2015*). The synapse-specific expression of Syn isoforms (*Bragina et al., 2010*; *Patton et al., 2016*; *Wei et al., 2011*) contributes to the diversity of STP profiles (*Feliciano et al., 2017*; *Gitler et al., 2004*; *Kielland et al., 2006*; *Song and Augustine, 2016*) and determines the inhibitory-excitatory balance in cortical and hippocampal networks (*Fassio et al., 2011*; *Ketzef and Gitler, 2014*). Here, we show that MLIs, regardless of their identity, receive four functionally distinct types of synapses from GCs. Each class of connection differentially drives the timing of MLI recruitment characterized by the first-spike latency. Differences in STP across GC-MLI connections are underlain by heterogeneous expression of Syn II at GC-MLI synapses. Functional studies using wild-type (WT) and Syn II knockout (KO) mice demonstrate that Syn II determines the profile of STP and the first-spike latency in MLI. Our observation that single MLIs receive GC inputs with distinct molecular and functional properties suggests that the temporal coding of GC activity by MLIs and thereby the inhibitory control over the cerebellar output via PCs depends on GC subtypes recruited by a given sensorimotor input.

## Results

### Functional heterogeneity at unitary GC-MLI synapses during high-frequency stimulations

In order to study how information from single GC inputs is encoded by MLIs, we measured STP at unitary GC-MLI synapses in acute parasagittal slices by minimal electrical stimulation of PFs (10 pulses at 100 Hz), in the direct vicinity of the dendritic tree of a recorded MLI (*Malagon et al., 2016*; *Miki et al., 2016*). MLIs localized in the vermis (lobules IV-VI) were recorded in whole-cell voltage-clamp configuration and loaded with Atto-594 (*n* = 49) to visualize their dendritic tree and their morphology (*Figure 1—figure supplement 1A*). Using two-photon microscopy, the stimulation pipette was visually positioned above an isolated dendrite. The stimulating currents were carefully adjusted to stimulate only a single synaptic contac (see Materials and methods and *Figure 1—figure supplement 1C–E*). In 28 MLIs, we recorded synaptic responses from at least 2 GCs. STP profiles were highly heterogeneous across unitary GC-MLI contacts (*n* = 96) including unitary connections contacting the same MLI (*Figure 1*). To classify the STP profiles, we used principal component analysis

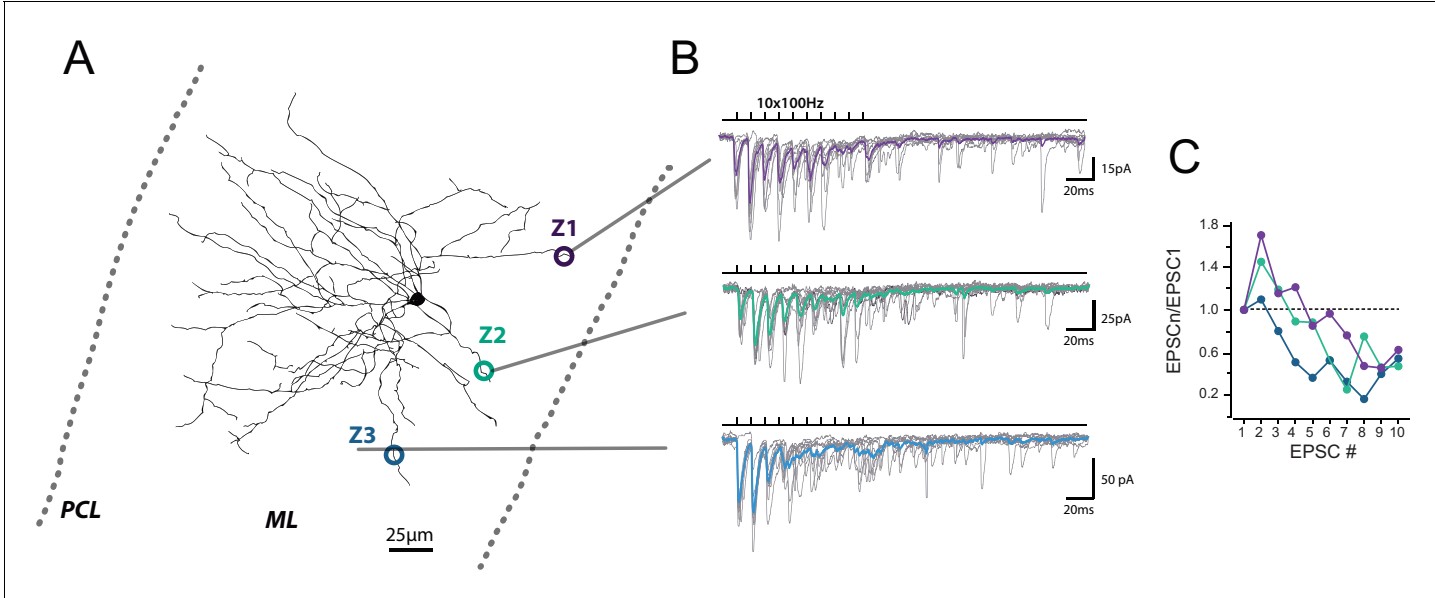

**Figure 1.** Heterogeneous profile of STP at unitary GC-MLI synapses. (A) Typical experiment showing the profile of STP during 100 trains at three unitary inputs recruited by local stimulation of PF at two different locations (Z1 to Z3). Figure shows post-hoc reconstruction of a recorded MLI. The left and right dashed lines represent the location of the Purkinje cell layer (PCL) and the pia, respectively. (B) Superimposed traces correspond to EPSCs recorded during trains of 10 stimuli at 100 Hz after minimal stimulation at Z1, Z2 and Z3 locations. Averaged traces from 10 successive stimulations are represented in purple (Z1), green (Z2) and blue (Z3). (C) Corresponding EPSC charges versus stimulus number at Z1, Z2 and Z3 locations.

DOI: https://doi.org/10.7554/eLife.41586.002

The following figure supplement is available for figure 1:

**Figure supplement 1.** Minimal stimulation protocol.

DOI: https://doi.org/10.7554/eLife.41586.003

(PCA) on the averaged and normalized synaptic charges in trains of EPSCs followed by a *k*-means clustering analysis (*Figure 2—figure supplements 1* and *2*). We identified four clusters that characterize STP at GC synapses (*Figure 2A–B*). The profiles differ by: (i) the quantity of neurotransmitter released at the first stimuli (*Figure 2C, left*), (ii) paired-pulse plasticity (*Figure 2C, right*), (iii) the STP profiles during the first four EPSCs, and (iv) the ability to sustain glutamate release after the fourth stimuli. Because high-frequency stimulation can change the excitability of PFs, it should be ensured that no additional PFs are recruited during the train. Given precautions made to select the lowest stimulation intensity (see Materials and methods), such recruitment occurs randomly during the train. Recruitment of additional PFs may strongly affect the profile of STP from one train to another during successive stimulations and skew the classification of inputs. We checked whether the profile of STP was conserved during 10 successive 100 Hz trains. The systematic narrow clustering of 10 recordings belonging to the same series of stimulations in the cloud of point of PCA transformation (*Figure 2—figure supplement 2A,B*) clearly indicated that the profiles of STP were conserved from one train to another during successive stimulations of the same synaptic contact. Recruitment of additional PFs might also depend on the intensity of stimulation. However, the lack of correlation between the charge of the first EPSCs, PPR, PC1 or PC2 and the stimulation current intensity indicated that STP profiles were not affected by this parameter (*Figure 2—figure supplement 2C*). Choosing intensities just above the threshold could also induce failure of fiber recruitment rather than presynaptic mechanisms. However, the classification of inputs using PCA transformation followed by *k*-mean clustering analysis was weakly affected when EPSCs at the first stimulus were excluded from the dataset (*Figure 2—figure supplement 3*). Hence, putative experimental errors due to unexpected changes in PF excitability did not influence the overall classification of inputs.

Among the four groups identified, only C1 connections exhibited depression of glutamate release after the second pulse, while C2 and C3 connections exhibited facilitation (*Figure 2D*). C2 and C3 connections differed in their responses after the fourth stimulus: while C3 connections sustained

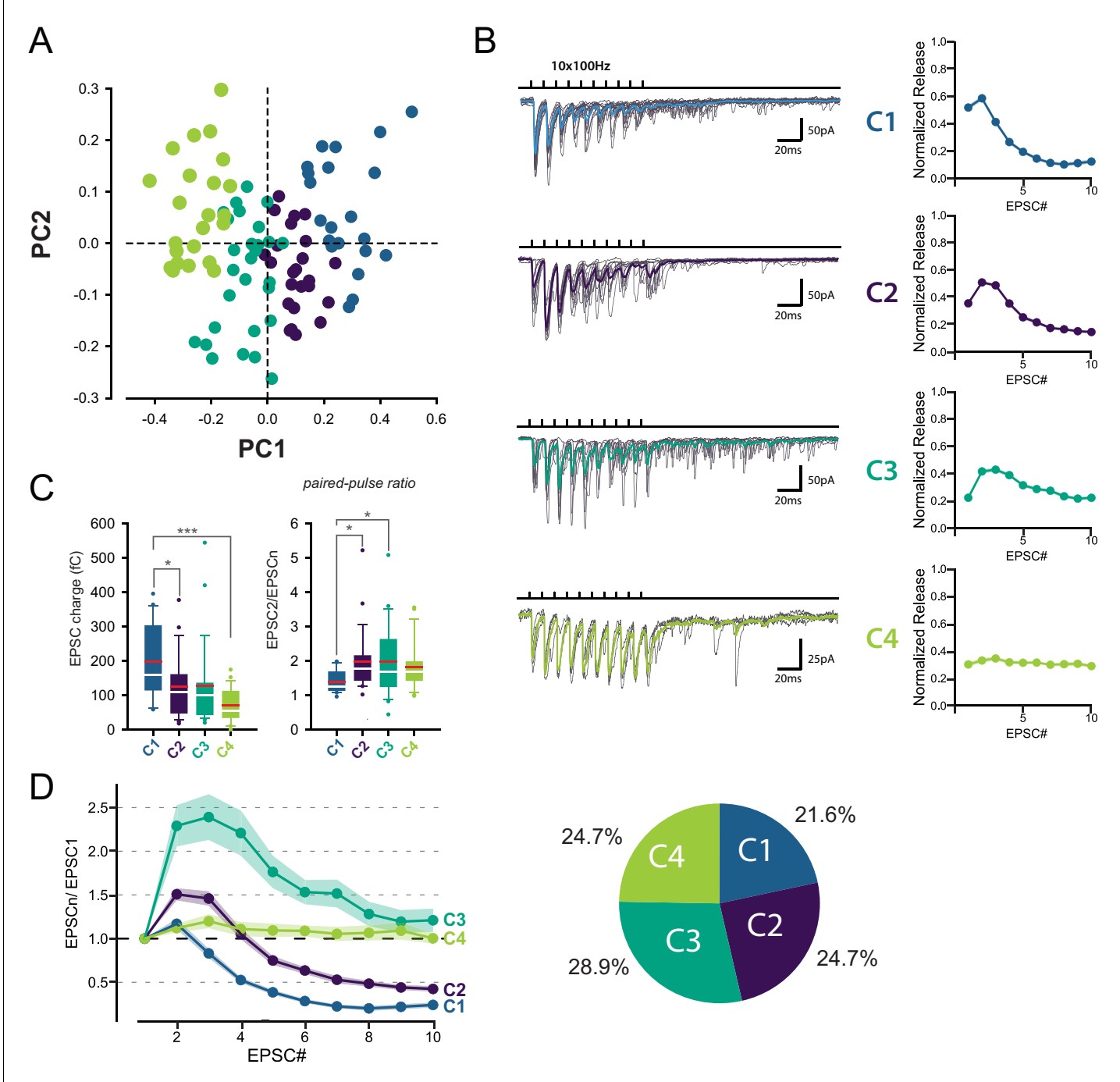

**Figure 2.** Identification of 4 classes of MC-MLI synapses using PCA followed by *k*-means clustering analysis of EPSC properties during high-frequency stimulation. (**A**) PCA transformation of GC-MLI STP profiles. Scatter plot of the first two principal components (PC1, PC2) obtained by analyzing EPSC properties during 100 Hz trains at numerous unitary GC-MLI synapses (*n* = 96). The first two components explain 69.4% of the total variance of STP. Synapses with negative PC1 values sustain glutamate release during the 10 EPSCs of the burst while PC1 with positive values synapses are depressing synapses. Positive PC2 synapses are depressing synapses while negative PC2 synapses are facilitating during EPSC #2 and EPSC #3. (**B**) Representative traces of the four classes of inputs (C1 to C4) determined by *k*-means clustering analysis during ten minimal stimulations of unitary inputs at 100 Hz. The corresponding values of averaged EPSC amplitudes plotted versus the stimulus number and normalized again the vector space model (see method) were displayed on the right panels. (**C**) Box plots of the charge of EPSC recorded at the first stimulus (*left panel*) and the paired pulse ratio (PPR) (*right panel*) according to the four categories of input. EPSC charges: C1 = 198.78 fC±23.46 fC, C2 = 124.96 fC±18.78 fC, C3 = 126.52 fC±22.10 fC, C4 = 74.41 fC±9.48 fC. PPR: C1 = 1.39 ± 0.07, C2 = 1.97 ± 0.19, C3 = 1.98 ± 0.19, C4 = 1.82 ± 0.14. Red bars refer to means and white bars to medians. Multiple comparisons were performed using one-way ANOVAs with Tukey *post hoc* tests. Only statistically significant differences between categories were

*Figure 2 continued on next page*

*Figure 2 continued*

shown above box plots (**D**) Mean values of normalized EPSC amplitudes during 100 Hz train according to the four categories of inputs. The circular diagram represents the relative proportion of each category of input from 96 unitary GC-MLI synapses.

DOI: https://doi.org/10.7554/eLife.41586.004

The following figure supplements are available for figure 2:

**Figure supplement 1.** Details on Principal Component Analysis and *k*-Mean clustering parameters of minimal stimulation GC-MLI STP data.

DOI: https://doi.org/10.7554/eLife.41586.005

**Figure supplement 2.** Robustness of intrinsic STP evoked by 100 Hz train at unitary GC-MLI synapses.

DOI: https://doi.org/10.7554/eLife.41586.006

**Figure supplement 3.** The classification of GC-MLI synapses using PCA transformation followed by *k*-means clustering analysis was weakly impacted by the first response.

DOI: https://doi.org/10.7554/eLife.41586.007

release during the entire train, synaptic transmission at C2 connections depressed after the fourth stimuli. On the other hand, C4 connections were characterized by small but stable EPSCs (*Figure 2C*) suggesting that they correspond to boutons releasing one quanta per stimulation (*Bender et al., 2009*; *Nahir and Jahr, 2013*). MLI subtypes were identified in half of the recorded cells based on the presence or not of a basket in the Purkinje cell layer (*n* = 13 for BCs, *n* = 12 for SCs and *n* = 24 for non-identified MLI). Both MLI subtypes were contacted by functionally distinct GC inputs. Except for C3 connections (facilitating profiles) absent in BCs, all classes of GC inputs were found on SCs and BCs (*Figure 3*). The lack of target-cell dependence of STP at GC-MLI synapses was also confirmed using compound stimulations of GCs in the granule cell layer (*Figure 3—figure supplement 1*). The strict segregation of MLIs in two subclasses was challenged by several authors and studies (*Jörntell et al., 2010*; *Rakic, 1972*; *Sotelo, 2015*; *Sultan and Bower, 1998*). Nevertheless, morphological features of MLIs were found to be related to the position of each MLI's soma in the deepness of the molecular layer (*Sultan and Bower, 1998*). Our analyses failed to establish correlations between STP profiles (evaluated by PC1 or PC2) and the position of MLIs' soma (*Figure 3—figure supplement 2A*) or synaptic inputs (*Figure 3—figure supplement 2B*) in the molecular layer. Finally, since dendritic integration of excitatory responses in MLI may be influenced by the distance of the synapses from the soma (*Tran-Van-Minh et al., 2016*) we also checked whether the distances of excitatory inputs from MLI's soma were correlated with STP parameters. Again, our analyses failed to establish a correlation between those parameters (*Figure 3—figure supplement 2C*). Our results indicate that the different classes of inputs recruited by minimal stimulations were randomly distributed within the molecular layer.

## Syn II is heterogeneously expressed across GC-MLI presynaptic terminals

Next, we aimed to uncover the role of Syn in functional heterogeneity. We first studied the presence of Syn I and Syn II in GCs boutons by immunohistochemistry using VGluT1 as specific marker of GC presynaptic terminals (*Hioki et al., 2003*; *Zander et al., 2010*). Triple staining of cerebellar sections from P20 ~ P22 CD1 mice (*N* = 4) revealed systematic overlap of VGluT1 with Syn I, but not with Syn II (*Figure 4A,B*). In presynaptic terminals, VGluT1 and Syn are supposed to be partially colocalized because VGluT1 is localized on synaptic vesicles while Syn are both cytosolic and associated with synaptic vesicles (*Cesca et al., 2010*). Quantitative analysis revealed a higher correlation between the fluorescence intensity of Syn I and VGluT1 than between the fluorescence intensity of VGlut1 and Syn II ($R_{Syn\ I}$ =0.584 +/- 0.057, $R_{Syn\ II}$ = 0.435 +/- 0.075, paired *t*-test: p<0.001; *n* = 52). Our results suggest that Syn I is present in all GC terminals while Syn II is restricted to a subpopulation of GC boutons.

Since most of GC synapses stained by VGluT1 actually correspond to GC-PC synapses, we could not exclude that Syn II is homogeneously expressed in GC-MLI synapses. We then performed pre-embedding immunogold labeling (*Figure 4C*) of Syn I and Syn II in parasagittal cerebellar sections. Asymmetrical GC-MLI synapses in the upper part of the molecular layer were identified by the presence of mitochondria within the postsynaptic compartment (*Palay and Chan-Palay, 1974*). Immunogold labeling confirmed the ubiquitous presence of Syn I in all GC-MLI and presence of Syn II in only 56% of GC terminals contacting MLI (*Figure 4D*).

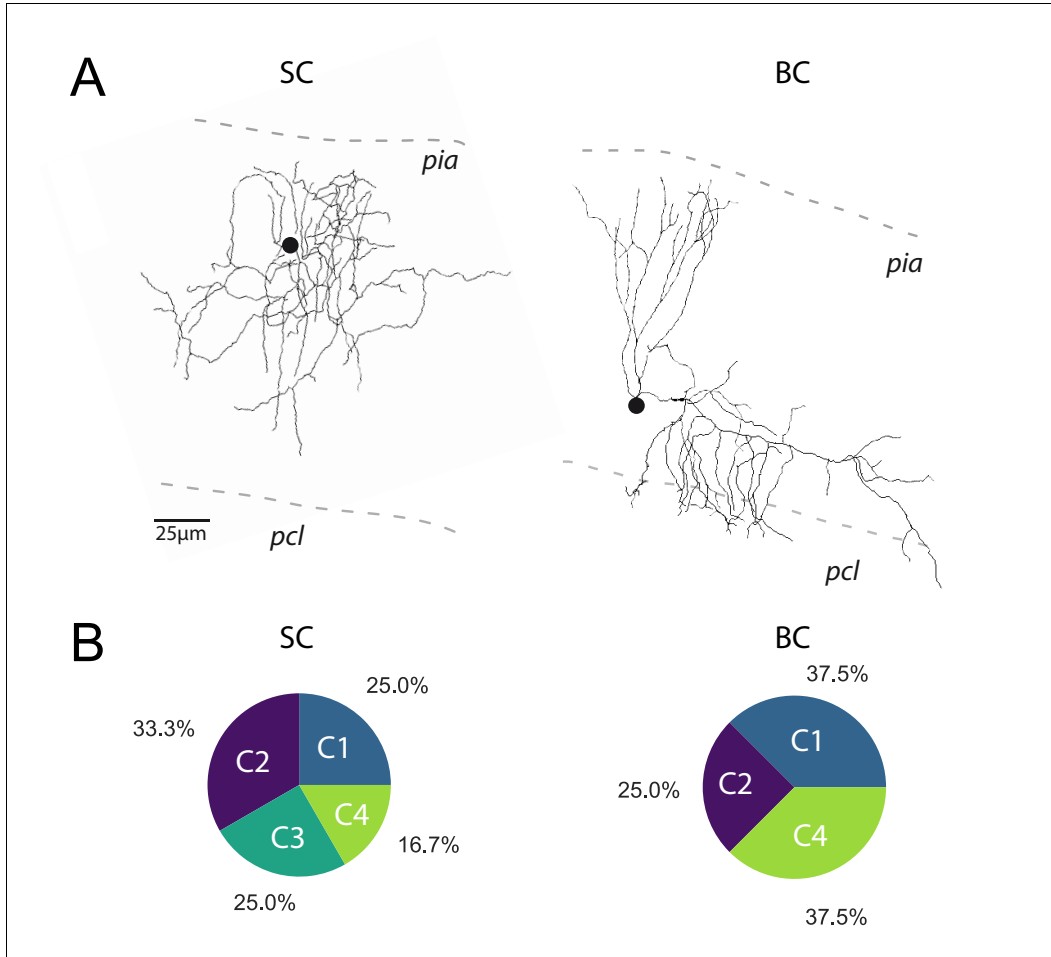

**Figure 3.** The profile of STP is not determined by the target cell. (A) *Post-hoc* reconstruction of 2 recorded MLI using a two-photon microscope. SCs were identified by the absence of neuronal process reaching the PCL (left MLI) and by the absence of cut processes (transection of neuronal processes could be clearly identified by swelling at the tip end portion of processes). At the opposite, BCs were identified by the presence of processes entering in the PCL (right MLI). (B) Circular diagrams of the relative proportion of each category of input (determined by *k*-means clustering analysis during ten minimal stimulation of GC unitary inputs at 100 Hz) contacting BCs and SCs.
DOI: https://doi.org/10.7554/eLife.41586.008

The following figure supplements are available for figure 3:

**Figure supplement 1.** STP profile at compound GC-MLI synapses is not determined by the target cell upon.
DOI: https://doi.org/10.7554/eLife.41586.009

**Figure supplement 2.** STP is not determined by the position of MLI in the molecular layer or by the position of inputs in MLI dendritic trees.
DOI: https://doi.org/10.7554/eLife.41586.010

## Heterogeneous expression of Syn II generates diversities in the profile of STP at unitary GC-MLI synapses

The heterogeneous expression of Syn II in GC terminals may contribute to the diversity of STP profiles across unitary GC-MLI synapses. To test this possibility, we reinvestigated STP diversity at unitary GC-MLI synapses in Syn II KO mice (*Figure 5A,B*). Absence of Syn II modified the responses of unitary GC-MLI synapses to 100 Hz stimulations. The mean EPSC charges of the first responses was strongly reduced in Syn II KO mice (*Figure 5B,C*). The percentage of failures at the first stimuli were increased in Syn II KO mice indicating that absence of Syn II decreased the probability of release ($p_r$) of fully-releasable synaptic vesicles (that is, vesicles released by a single action potential,

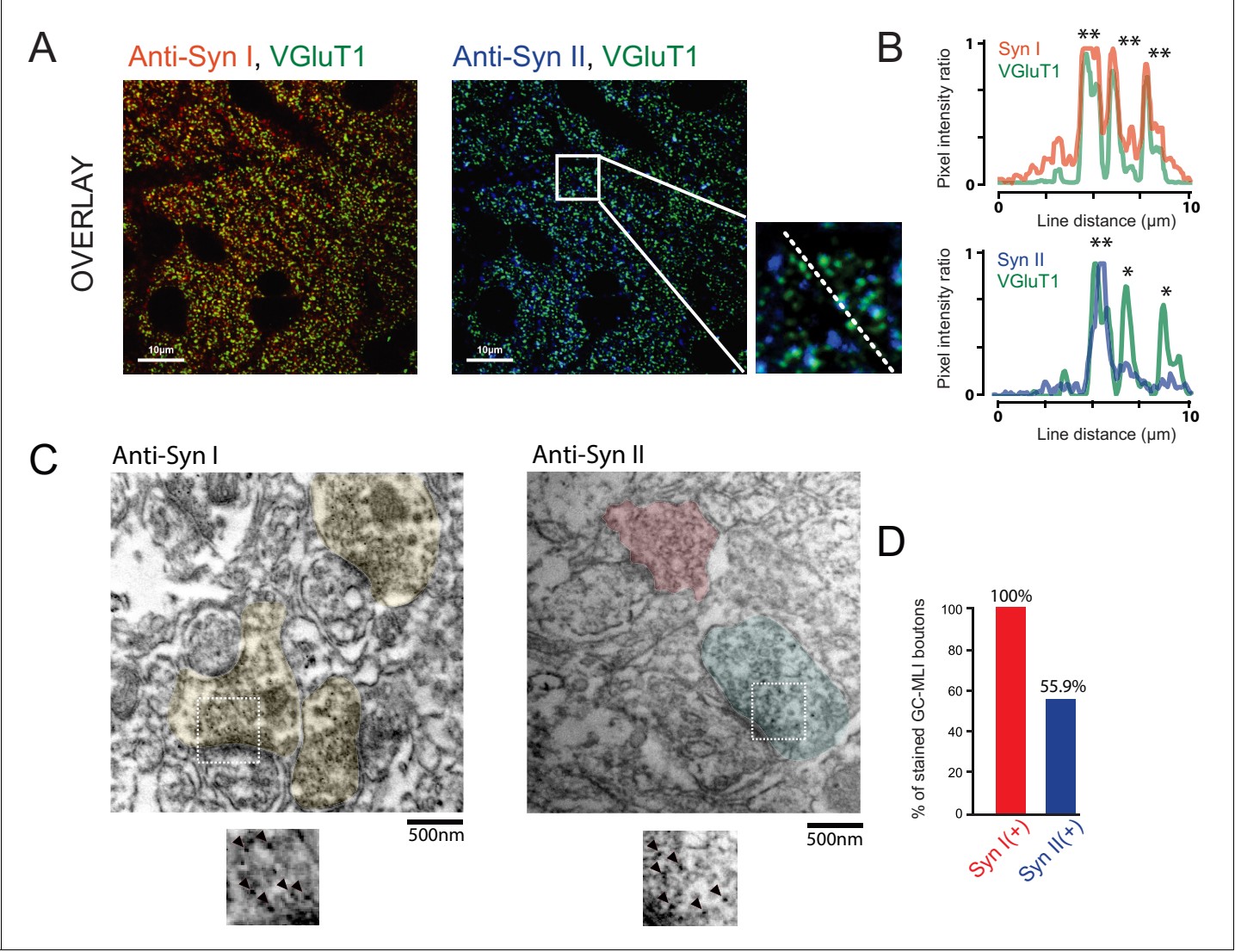

**Figure 4.** Heterogeneous expression of Syn II at GC-MLI synapses. (**A**) Representative merged images of VGluT1/Syn I immunostaining (*left image*, green and red puncta, respectively) or VGluT1/Syn II immunostaining (*right image*, green and blue puncta, respectively). The two merged imaged were captured in the molecular layer from the same parasagittal cerebellar section. (**B**) Profile plot (dashed line in *A*) showing the colocalization of VGluT1 with Syn I in the majority of VGluT1 puncta while there was only a partial colocalization of VGluT1 with Syn II in VGluT1 puncta. (**C**) Typical immunogold electron micrographs illustrating the ubiquitous expression of Syn I in GC boutons contacting MLIs (*left micrograph*) and the heterogeneous expression of Syn II in these boutons (*right micrograph*). GC boutons contacting MLIs were colorized. Insets corresponding to magnifications of areas delimited by white squares show details of immunogold staining. (**D**) Histogram of the percentage of GC-MLI synapse positive for Syn I (red bar) and Syn II (blue bar).

DOI: https://doi.org/10.7554/eLife.41586.011

*Doussau et al., 2017*) (*Figure 5D*). The paired-pulse ratio was significantly increased in Syn II KO mice (mean/median PPR for WT = 1.1/1.1 ± 0.03, *n* = 96 and mean/median PPR for Syn II KO mice = 2.6/1.7 ± 0.08, *n* = 53, p<0.001, MWRST). We then analyzed STP profiles at unitary connections in Syn II KO as in *Figure 2* and plotted individual profiles against the first two dimensions of a PCA based on the PCA fit of WT data (Materials and methods) and examined the spread of Syn II KO individual GC-MLI STP observations (*Figure 5E*). The four profiles of STP found at unitary GC-MLI synapses in WT mice were also found at unitary GC-MLI synapses in Syn II KO mice. However, the distribution of the four classes was strongly skewed toward C3 and C4 profiles (85.8% of the connections, *n* = 33) in Syn II KO mice (*Figure 5E,F*) while C1 and C2 connections almost disappeared (C1 connection 3.6%, C2 connections 10.7%) (*Figure 5E*). These results suggest Syn II lead

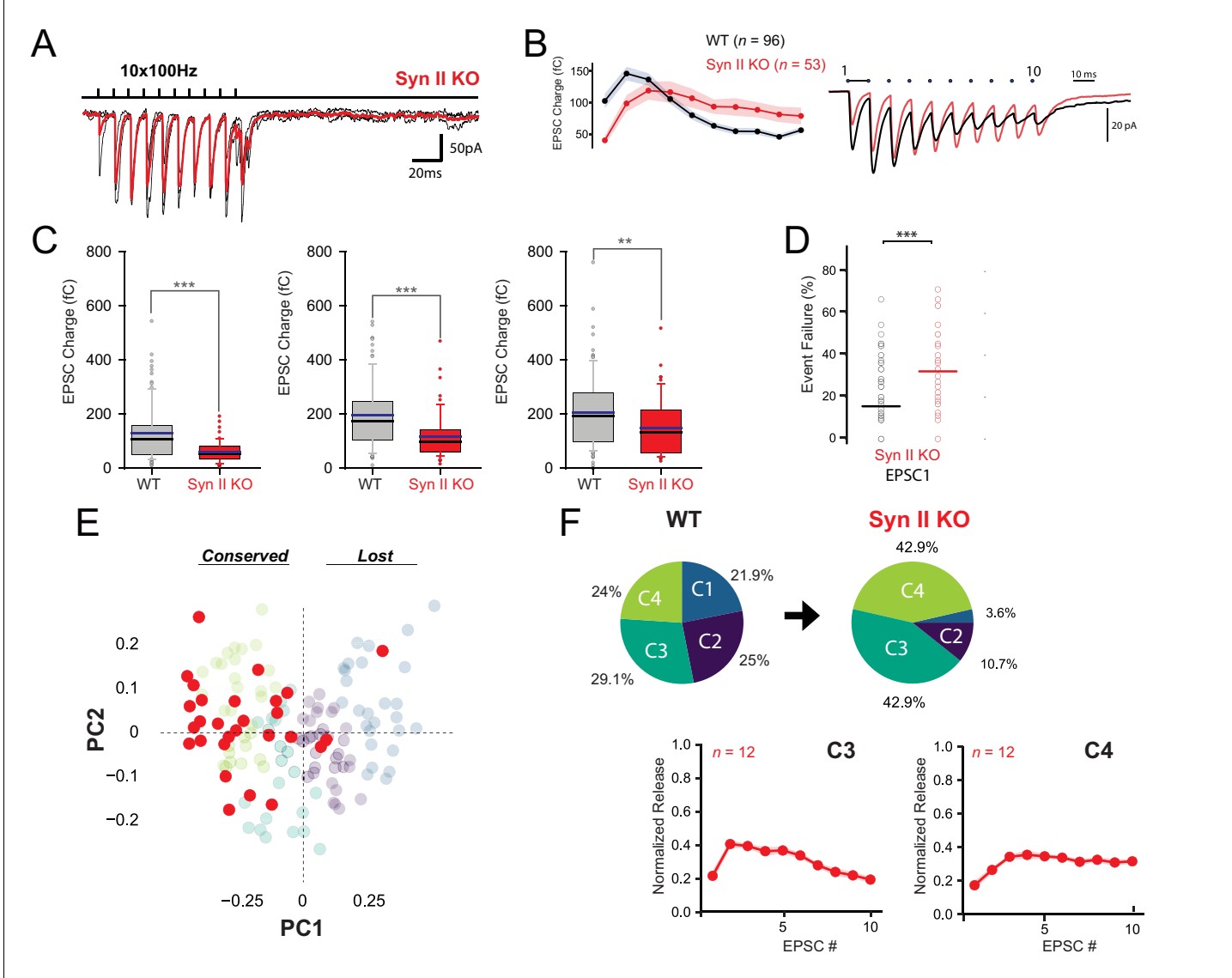

**Figure 5.** Genetic deletion of Syn II induces a partial loss of functional variability at GC-MLI synapses. (**A**) Representative traces of EPSCs from 10 successive trains at 100 Hz recorded at unitary GC-MLI synapse from Syn II KO mice (black traces). Averaged trace is in red. Unitary synapses were stimulated using minimal electrical stimulation. (**B**) *left*, Mean values of EPSC1, charges elicited by train of stimulation at 100 Hz recorded in WT and Syn II KO mice (grey and red points, respectively). *Right*, Corresponding traces recorded during these 100 Hz train in WT mice (black trace, mean trace from 102 recordings) and in Syn II KO mice (red trace, averaging from 33 recordings). The mean EPSC charges of the first responses was strongly reduced in Syn II KO mice (mean EPSC1 charge for WT: 128.11 fC ± 10.51 fC, $n = 96$, mean EPSC1 charge for Syn II KO mice: 60.03 fC ± 5.55 fC, $n = 53$, p<0.001, MWRST). (**C**) Box plots showing the values of EPSC charges at the first, second and third stimulus of 100 Hz train (left, middle and right graph respectively) in WT and Syn II KO mice. (**D**) Box plots showing the number of failures at the first stimulus in WT and Syn II KO mice. The percentage of failures at the first stimuli was increased in Syn II KO mice (mean failure rate EPSC1 in WT: 15.6% ± 1.6, Syn II KO mice: 32.3% ± 3.8). (**E**) Scatter plot of PCA1 and PCA2 obtained by analyzing EPSC properties during 100 Hz train in WT mice (gray point, same dataset as in *Figure 2A*) and Syn II KO mice (red points). (**F**) Pie chart of *k*-means clustering analysis clusters obtained in WT (same dataset than in *Figure 2C*) and Syn II KO mice. Note the near complete disappearance of C1 and C2 connections in Syn II KO mice. The profiles of EPSC charges during 100 Hz train for C3 and C4 connections were identical between WT and Syn II KO mice (*bottom graphs*), indicating that the genetic deletion of Syn II did not impair the functioning of these two classes of GC-MLI synapses.

DOI: https://doi.org/10.7554/eLife.41586.012

to STP profiles corresponding to C1 and C2, whereas GC boutons displaying C3 and C4 profiles are devoid of Syn II.

We next studied the subcellular localization of synaptic vesicles at GC-MLI synapses from WT and Syn II KO mice using transmission electron microscopy (*Figure 6A*). Morphometric analysis revealed that the absence of Syn II reduced significantly the number of docked synaptic vesicles and the length of the active zone (*Figure 6B*) without affecting the positive correlation between the length of the active zone and the number of docked synaptic (*Figure 6C*).

Altogether, our results suggest that the presence of Syn II positively regulate the number of docked synaptic vesicle and the $p_r$ of fully-releasable vesicles, thus enhancing the release glutamate at the onset of burst firing.

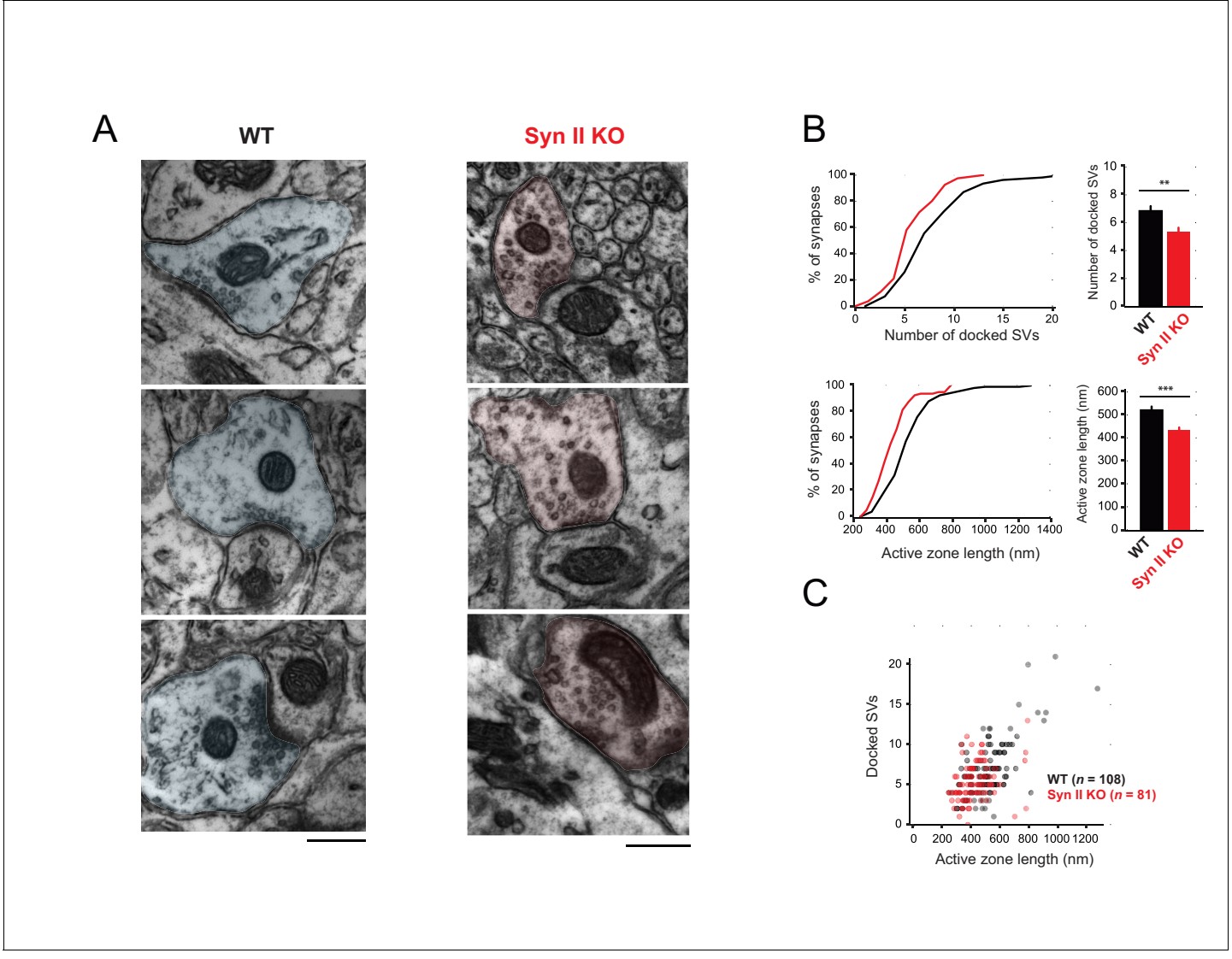

**Figure 6.** Genetic deletion of Syn II reduces the heterogeneity of ultrastructural profiles of presynaptic terminals at GC-MLI synapses. (**A**) Representative micrographs of GC-MLI synapses captured in the upper part of the molecular layer of cerebellar parasagittal sections from WT and Syn II KO mice. (**B**) *Upper panels*, Cumulative distribution (*left panel*) and mean values of the number of docked SVs at GC-MLI synapses from WT and Syn II KO mice (black line/bar and red line/bar respectively). *Lower panels*, similar representations for the active zone length. Absence of Syn II reduced significantly the number of docked synaptic vesicles and the length of the active zone (mean number of docked synaptic vesicles: WT 6.83 ± 0.35, n = 108; Syn II KO 5.36 ± 0.28, n = 81, mean length of active zone: WT 521.3 nm ± 15.2, n = 108; Syn II KO 432.2 nm ± 13.2, n = 81). (**C**) Scatter plot of the number of docked synaptic vesicles (SVs) versus the active zone (AZ) length from dataset obtained in B. Genetic deletion of Syn II led to a specific loss of GC bouton endowed with both a long active zone (>800 nm) and high number of docked synaptic vesicles (>15 synaptic vesicles).
DOI: https://doi.org/10.7554/eLife.41586.013

## Diversity of STP profile at GC-MLI connections extends the coding range of MLI

The STP profile shapes the spike output pattern of MLIs following compound stimulation of GCs or PFs (*Bao et al., 2010*; *Carter and Regehr, 2000*). This suggests that each class of GC-MLI synapse should influence the MLI spike output pattern specifically. To address this hypothesis, we set out to correlate STP of specific GC units with the spike output pattern of the targeted MLI. We recorded the spike output pattern of MLIs in loose-patch configuration following photostimulation of unitary GC inputs by caged glutamate (Materials and methods and *Figure 7—figure supplement 1*). Photostimulation of individual GCs increased the MLI firing rate confirming that sufficient glutamate was released by unitary GC boutons during high-frequency stimulation to produce spikes in MLIs (*Barbour, 1993*; *Carter and Regehr, 2002*) (*Figure 7A*). Photostimulations produced burst in GCs with reproducible parameters (*Figure 7—figure supplement 1*) and were followed by an increase in MLI firing rate (mean baseline frequency: 12.75 ± 5 Hz; peak of acceleration: 33.7 ± 17 Hz, *n* = 32). Subsequently, EPSCs were recorded upon photostimulation of the same unitary GC-MLI synapse using whole-cell configuration (*Figure 7A*). Photostimulations yielded heterogeneous profile of STP. PCA followed by *k*-means clustering analysis revealed three distinct STP profiles (C1', C2' and C3' connections different from C1 to C4 connections since the parameters of bursts elicited in GCs using minimal electrical stimulations and photostimulations are different) that differed by their time course and amplitude (*Figure 7B–C*). C1' connections with positive PC1 values were characterized by large responses that peaked at the onset of GC bursts and then rapidly depressed (phasic profile) (averaged EPSC peak charge: −232.8 fC ± 55.2 fC reached at 33.2 ms ± 11.3, *n* = 14). C2' connections with low PC1 values were also characterized by a phasic profile, but EPSCs have smaller amplitudes (averaged EPSC peak charge: −121.3 fC ± 21.6 fC reached at 59.7 ms ± 10.3, *n* = 30) than C1' synapses. C3' connections were characterized by smaller responses that peaked with longer delays than the ones of C1' or C2' synapses (EPSC peak charge: −133.4 fC ± 21.9 fC reached at 84.4 ms ± 5.4, *n* = 18).

In 12 unitary connections out of 62, we could correlate the spike output pattern recorded in loose-patch configuration with the STP profile recorded in whole-cell configuration (*Figure 7D*). We measured the time separating the onset of photostimulation and the time the recorded MLI is firing is at its maximum frequency (labeled as Delay to frequency peak in *Figure 7D*). Photostimulation of C1' connections accelerated MLI firing rate with a very short delay (<50 ms) compared to C2' and C3' connections (delay > 60 ms) (*Figure 7D*). Our analysis also showed a clear relationship between PC1 and the peak frequency (Pearson coefficient, R = 0.7, p=0,008, *n* = 13). This indicates that a specific STP profiles determines the delay to the first spike in MLI in response to GC photostimulation. These results suggest that the behavior of glutamate release at GC-MLI synapses during high-frequency stimulations is a major determinant of the coding of sensorimotor inputs through the FFI pathway.

In some experiments, after photostimulation of 2–3 GCs contacting the same MLI, we could perform *post-hoc* reconstruction of the recorded cell. Similarly to what was observed with minimal stimulation experiments (*Figure 3*), we found that identified SCs and BCs both received inputs of different classes upon photostimulation of individual GCs (*Figure 7*, *Figure 7—figure supplement 2*). These results confirmed the lack of target-cell-dependent STP at the level of individual connection at GC-MLI synapses.

We next tested how Syn II deficiency affects the correlation between STP profiles and firing pattern at GC-MLI synapses, using Syn II KO mice (*n* = 27). Accordingly, photostimulation of single GCs with RuBi-glutamate were performed in Syn II KO mice as described above for WT mice (*Figure 7* and *Figure 7—figure supplement 1*); evoked spikes and synaptic responses were recorded in MLIs using loose-patch and whole cell recordings. PCA followed by *k*-means clustering analysis showed that genetic deletion of Syn II almost abolished the presence of C1' connections and nearly doubled the percentage of C3' connections (*Figure 8A–D*). Moreover, the delay to the first spike significantly increased in Syn II KO mice (*Figure 8E,F*), probably due to a lower initial $p_r$ at GC-MLI synapses devoid of Syn II (*Figure 5A,B*). Our results reveal that Syn II is a major determinant of burst coding at the GC-MLI synapses. Synapse-specific expression of Syn II diversifies the profile of excitatory drives on MLIs and expands the coding range in the FFI pathway.

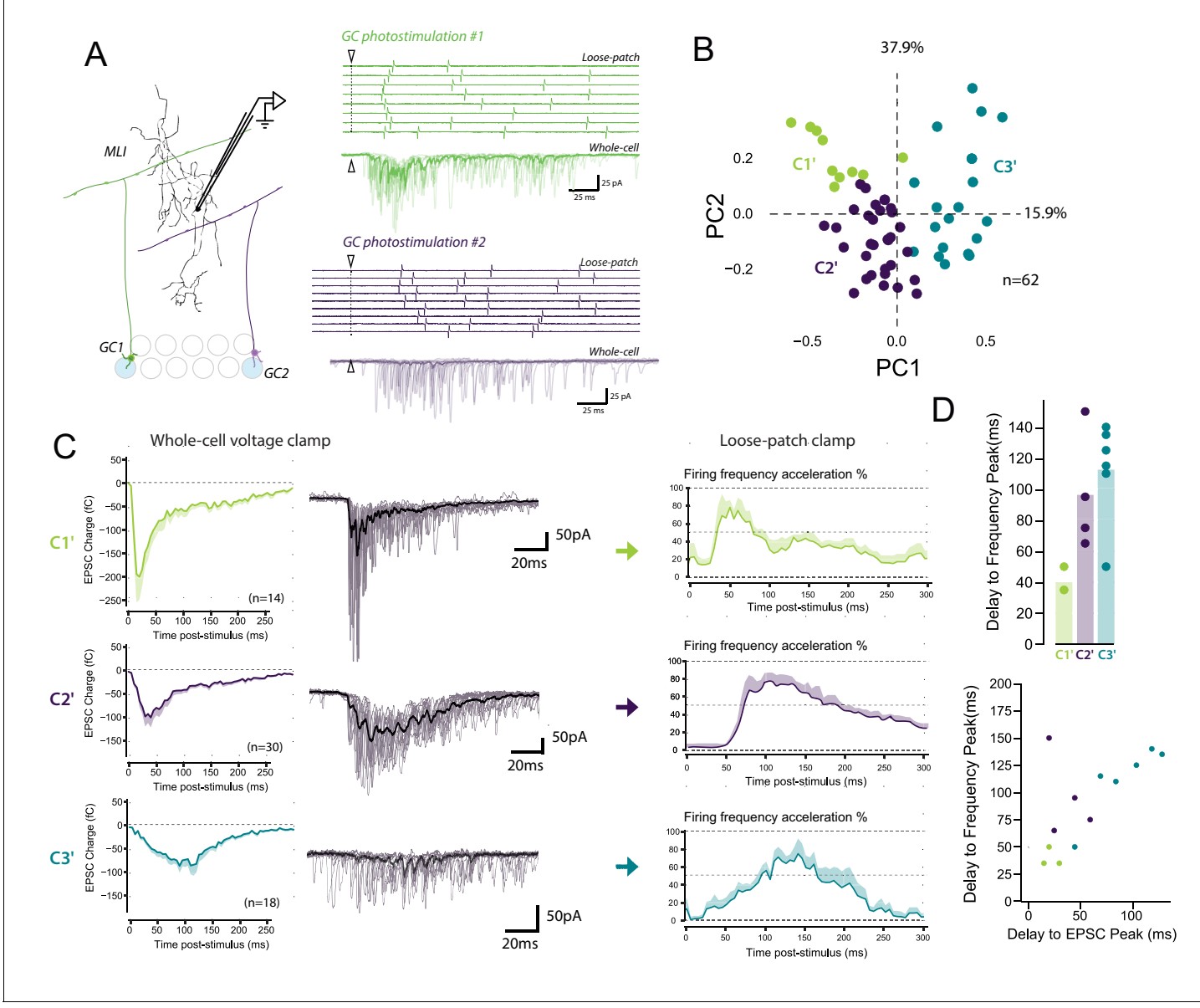

**Figure 7.** The diverse profiles of STP of GC boutons differentially shape the spike output pattern in the target MLI. (**A**) *Left panel*, schematic representing the design of photostimulation in the granule cell layer. Open circles represent the sites where RuBi-glutamate was uncaged and the blue circles represent the locations where photostimulation elicited responses in the recorded MLI. In this example, 2 GCs localized at distal positions in the granule cell layer contact the recorded MLI. *Right panels*, representative experiment showing the spike output pattern recorded in loose-patch configuration and EPSCs recording in whole cell configuration in the same MLI following photostimulation of two different locations in the granule cell layer. The white arrowheads and dashed lines represent the onset of photostimulation. Note that the onset of firing is time-locked to the first peak of EPSC charge for photostimulations in location #1 (*upper panels*) while the onset of firing was more variable for photostimulation in location #2 (*lower panels*). (**B**) PCA transformation of the evoked charge time course for 63 unitary contacts (see Materials and methods). The EPSC bursts could be differentiated depending on their tonic or phasic component into three different clusters using *k*-Means clustering analysis. (**C**) Representative traces of EPSCs and of the corresponding firing profiles recorded in singles MLI following stimulation of C1', C2' and C3' connections. (**D**) The delays separating the onset of photostimulation and the time the recorded MLIs were firing at their maximum frequency (referred as delay to frequency peak, recorded in loose-patch configuration) and the delays separating the onset of photostimulation and the time EPSCs are reaching their maximum value (referred as delay to EPSC peak, recorded in voltage-clamp configuration) were measured for 12 GC-MLI synapses for which we were able to correlate the spike output pattern with STP profile (C1', C2' or C3'). *Upper panel*, the bar plot shows that stimulation of C1' connections led to faster accelerations of MLI firing rate than stimulation of C2' and C3' connections. *Lower panel*, the scatter plot shows a clear correlation between the delay to EPSC peak and the delay to frequency peak recorded. Each experimental point was associated with its STP profile by using the same color code for categories than in the upper graph.

DOI: https://doi.org/10.7554/eLife.41586.014

*Figure 7 continued on next page*

*Figure 7 continued*

The following figure supplements are available for figure 7:

**Figure supplement 1.** RuBi-glutamate uncaging induced reproducible high-frequency bursts in GCs.

DOI: https://doi.org/10.7554/eLife.41586.015

**Figure supplement 2.** The profile of STP is not determined by the target cell in photostimulation experiments.

DOI: https://doi.org/10.7554/eLife.41586.016

## Discussion

By combining molecular, ultrastructural and functional studies, we show that the firing pattern of MLIs is driven by distinct GC inputs that show distinct profile of STP. This functional heterogeneity caused, at least in part, by synapses-specific expression of Syn II expands the coding range of MLIs.

### MLI subtype does not determine the profile of STP at unitary GC-MLI synapse

The rules governing diversity in presynaptic release properties have been extensively studied in neocortical or hippocampal circuits. In most of cases, synaptic efficacy among boutons issued from a single axon varies with the identity of the postsynaptic cell (*Blackman et al., 2013*; *Markram et al., 1998*). Target-cell-dependent heterogeneity relies on differences in the probability of release (*Koester and Johnston, 2005*), responsiveness to neuromodulators (*Buchanan et al., 2012*; *Delaney and Jahr, 2002*; *Pelkey et al., 2006*; *Scanziani et al., 1998*) or the ability to co-release GABA and glutamate (*Galván and Gutiérrez, 2017*). Target-cell-dependent STP has also been described at cerebellar GC-MLI synapses by using stimulation of beam of PFs or clusters of GC somata; upon high-frequency stimulation, compound synaptic responses exhibit a facilitating profile at GC-SC synapses whereas these responses depress at GC-BC synapses (*Bao et al., 2010*). At the opposite, our results show rather heterogeneous unitary inputs to BCs (contacted by three different classes of inputs) or SCs (contacted by four different classes of inputs) (*Figure 3*). Although the lack of the main facilitating class of input (C3) on BCs may explain why compound GC-BC synaptic responses depress during high-frequency activities, our results argue against target-cell dependency of STP at GC synapses. It is likely that in compound responses, input classes associated with a strong synaptic strength mask the presence and influence of weak inputs exhibiting other profiles. Anyhow, our work suggests that the excitatory drive to MLIs and the tuning of the FFI pathway are more complex than expected.

### Organization of synaptic diversity at unitary GC-MLI synapses

Heterogeneous expression and functions of Syn II in different cell types forming a same neural network have been reported previously (*Bragina et al., 2010*; *Feliciano et al., 2017*; *Gitler et al., 2004*; *Kielland et al., 2006*; *Patton et al., 2016*; *Wei et al., 2011*) but our findings bring the first evidence that expression of Syn II at a given connection can be heterogeneous. Syn II expression may be genetically determined at early developmental stages, leading to Syn II(+) and Syn II(-) subclones of GCs. It is interesting to note that clonally related GCs (that is, GCs issued from the same GC progenitors) stack their axons in a specific sub-layer in the molecular layer (*Espinosa and Luo, 2008*) suggesting that the presence or absence of Syn II may be organized in a beam-dependent way. Since beams of neighboring PFs are activated during sensory stimulations (*Wilms and Häusser, 2015*), recruitment of Syn II(+) or Syn II(-) connections may be related to the activation of a given sensorimotor task. Alternatively, Syn II targeting at individual PF boutons may be controlled by a complex interplay of mechanisms regulating the traffic of Syn in axons (*Gitler et al., 2004*) or organizing the assembly of the presynaptic active zone (*Owald and Sigrist, 2009*).

Presynaptic diversity also arises from other parameters that probably expand the range of synaptic behaviors across GC boutons. Calcium imaging performed on single PFs revealed that $Ca^{2+}$ dynamics and regulation of $Ca^{2+}$ influx by neuromodulators in synaptic varicosities from a same PF are highly heterogeneous (*Bouvier et al., 2016*; *Brenowitz and Regehr, 2007*; *Zhang and Linden, 2009*; *Zhang and Linden, 2012*). Also, local retrograde release of endocannabinoid by MLI dendrites can affect the functioning of subsets of GC boutons upon sustained activity of PF (*Beierlein and Regehr, 2006*; *Soler-Llavina and Sabatini, 2006*). Hence, functional heterogeneities

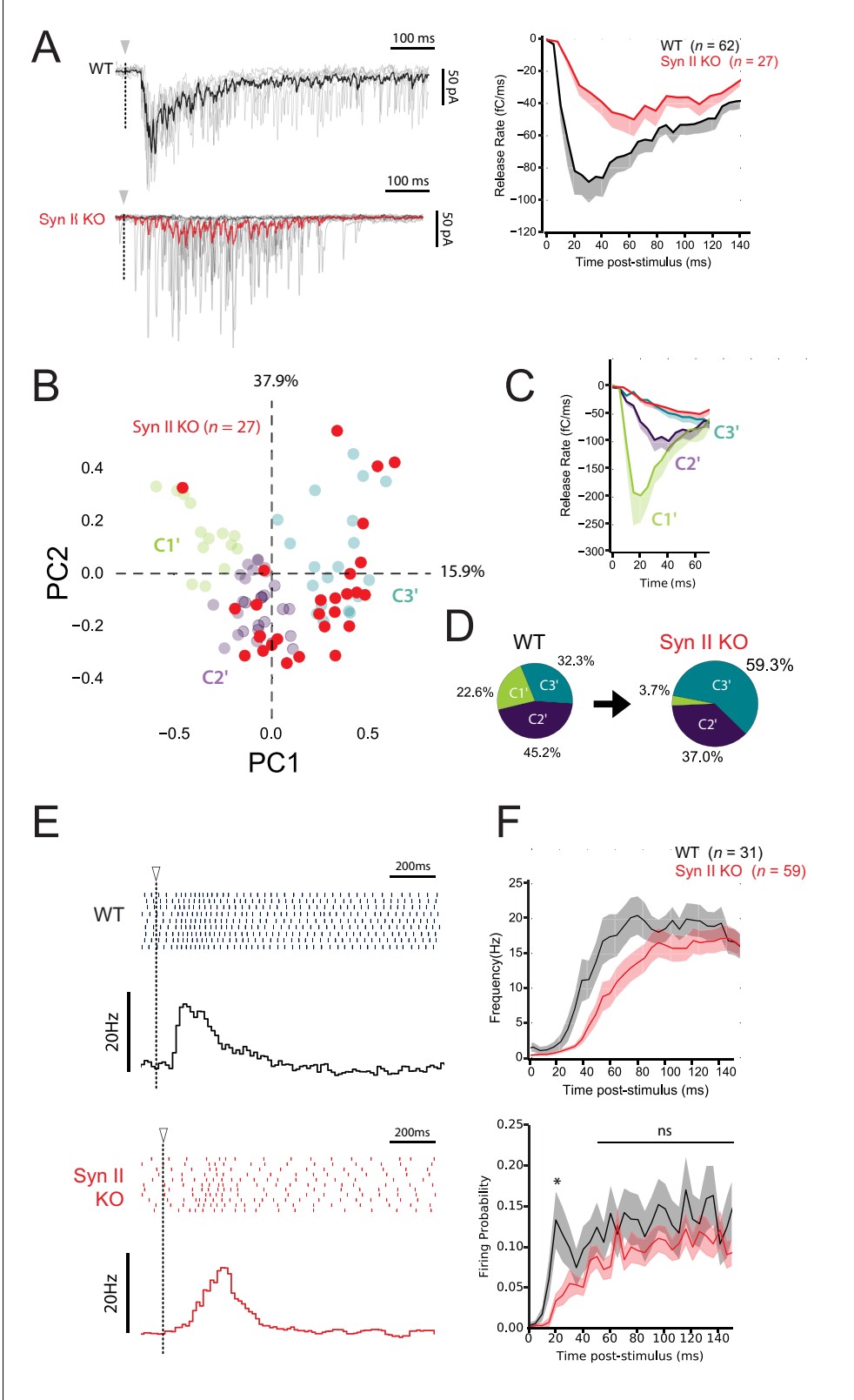

**Figure 8.** Synapse-specific expression of Syn II diversifies the profile of excitatory drives on MLIs and expands the coding range of MLIs. (**A**) *Left panels*, representative traces of superimposed EPSCs recorded in WT and Syn II KO mice after photostimulations in the GCL (same experimental design than in *Figure 7A*). *Right panel*, averaged values of EPSC charges versus the time following photorelease of RuBi-glutamate recorded in WT and Syn II KO

*Figure 8 continued on next page*

*Figure 8 continued*

mice (black and red traces respectively). Note the strong reduction in the peak of charge in Syn II KO mice. (**B**) PCA transformation of EPSC properties obtained in WT (green, purple and blue points, same dataset than in *Figure 7B*) and Syn II KO mice (red points). (**C**) Line plots display the normalized release time course from GC-MLI synapses belonging to clusters C1', C2', C3' (WT mice, same color code as in *B*) and GC-MLI synapses from Syn II KO mice (red line). (**D**) The pie chart shows a partial reduction of STP heterogeneity in Syn II KO condition with a strong reduction of phasic profiles (**C1'**). (**E**) Typical raster plots and peristimulus time histogram obtained in WT and Syn II KO mice following photostimulation of unitary GC-MLI synapses. The onsets of photostimulation are represented with white arrowheads and dashed lines. (**F**) Means values of the firing frequency (*upper graph*) and the firing probability (*lower graph*) of MLIs following photostimulation of unitary GC-MLI synapses in WT and Syn II KO mice (black and red lines, respectively).

DOI: https://doi.org/10.7554/eLife.41586.017

among GC terminals also originate from the history of firing of each GC. To summarize, the synaptic behavior of individual GC bouton may be tuned by an intermingled combination of factors including expression of Syn II, presynaptic receptor composition, presynaptic $Ca^{2+}$ dynamic, number of active release sites, retrograde signaling and history of firing.

## Control of glutamate release by Syn II in GC boutons

At GC synapses, releasable synaptic vesicles are segregated in two pools, one with fully releasable vesicles and a second one with reluctant vesicles, which are differentially poised for exocytosis (*Doussau et al., 2017*). The fully-releasable pool supports glutamate release during single action potentials while the reluctant pool is recruited only by stimuli elicited at high frequencies (*Doussau et al., 2017*). In Syn II KO mice, synaptic transmission is characterized by a defect in glutamate release by single action potentials and by a rapid recovery of synaptic transmission by 100 Hz stimuli. This suggests that a lack of Syn II impair $p_r$ of fully-releasable vesicles without affecting the recruitment of reluctant vesicles. Potentially, Syn II may act with several partners to control the recruitment of fully-releasable vesicles. In GC terminals, Munc13-3 has been involved in superpriming steps that tightly couple synaptic vesicles with P/Q-type $Ca^{2+}$ calcium channels (positional superpriming) or maturate the fusion machinery (molecular superpriming) (*Ishiyama et al., 2014*; *Kusch et al., 2018*; *Schmidt et al., 2013*). Munc13-3 may indirectly act with Rab3-interacting molecules (RIMs) which are well known organizers of calcium channel and synaptic vesicles in the active zone (*Südhof, 2013*). Since Syn II interacts with both Rab3 (*Giovedì et al., 2004*) and P/Q type calcium channels (*Medrihan et al., 2013*), it cannot be excluded that Munc13-3, Syn II, Rab3 and RIM act in concert to reduce the physical distance between fully-releasable vesicles and $Ca^{2+}$ channels. Alternatively, Syn II-Rab3-RIM complex may directly regulate the influx of $Ca^{2+}$ through strong inhibition of voltage-dependent inactivation of P/Q type $Ca^{2+}$-channels (*Hirano et al., 2017*; *Kintscher et al., 2013*).

## Physiological consequences

At the input stage of the cerebellar cortex, single GCs receive a combination of MF inputs coding for different modalities (*Arenz et al., 2008*; *Chadderton et al., 2014*). The diversity of STP profiles across MFs from different origins provide temporal signatures for each combination of MFs converging on a single GC thus enhancing pattern decorrelation of sensory inputs (*Chabrol et al., 2015*). Here, we show that temporal coding in GCs is later extended in the FFI pathway by an input-specific control of first-spike latency in MLIs. The combination of heterogeneous presynaptic behaviors at the successive stages of cerebellar computation leading to consecutive temporal signatures, should refine the salient feature of a given combination of MF inputs and ultimately should enhances the representation of sensory information by PCs.

Considering the importance of delay coding for internal models of motor adjustments (*Kennedy et al., 2014*; *Kistler et al., 2000*; *Mauk and Buonomano, 2004*; *Wolpert et al., 1998*) synapse-specific temporal coding may have essential consequences for learning and predictive functions in the cerebellum. Indeed, long-term potentiation (LTP) of GC-PC connections require the coincidence of strong PF and MLI activities onto the same PC (*Binda et al., 2016*). As exemplified by the rules governing the induction of associative GC-PC long-term depression triggered by coincident PF

and climbing fiber activation (*Suvrathan et al., 2016*), induction of associative GC-MLI LTP may depend on the time window separating excitatory and inhibitory inputs onto PCs. Heterogeneous profiles of STP at MF-GC synapses and GC-MLI synapses necessarily induce wide range of delays between the direct excitatory pathway and FFI at the level of PC synapses. Hence, the fine tuning of STP at the level of single cells may have fundamental importance for the induction of long-term plasticity and ultimately for motor learning.

# Materials and methods

## Key resources table

| Reagent type (species) or resource | Designation | Source or reference | Identifiers | Additional information |
|---|---|---|---|---|
| Genetic reagent (*M. musculus*) | Syn II KO mice (CD1 backround) | PMID 7777057 | | See Material and methods |
| Antibody | Rabbit anti-Syn I | Synaptic Systems | Cat# 106 104 | IHC (1:500) |
| Antibody | Monoclonal anti-Syn II Clone 27E3 | Synaptic Systems | Cat# 106 211 | IHC (1:500) |
| Antibody | Monoclonal anti-Syn II Clone 19.4 | Millipore | Cat# MABN 1573 | IHC (1:500) |
| Antibody | Guinea pig anti-VGluT1 | Synaptic Systems | Cat# 135 304 | IHC (1:600) |
| Antibody | Goat anti-Rabbit-Alexa 647 | Molecular Probes | Cat# A-21070 RRID:AB_2535731 | IHC (1:1000) |
| Antibody | Goat anti-Mice Alexa-488 | Molecular Probes | Cat# A-21141, RRID:AB_141626 | IHC (1:1000) |
| Antibody | Goat anti-Guinea Pig Alexa-555 | Molecular Probes | Cat# A-21435, RRID:AB_2535856 | IHC (1:1000) |
| Chemical compound, drug | Picrotoxin - GABA$_A$-R blocker | Abcam | Cat# Ab120315 | 100 µM in ACSF |
| Chemical compound, drug | CGP 52432 GABA$_B$-R blocker | Abcam | Cat# Ab120330 | 10 µM in ACSF |
| Chemical compound, drug | D-AP5 - NMDA-R blocker | Abcam | Cat# Ab120003 | 100 µM in ACSF |
| Chemical compound, drug | AM251 CB1-R blocker | Abcam | Cat# Ab120088 | 1 µM in ACSF |
| Chemical compound, drug | JNJ 16259685 – mGluR1 blocker | Tocris | Cat# 2333 | 2 µM in ACSF |
| Chemical compound, drug | Atto-594 | Sigma-Aldrich | Cat# 08637 | 50 µM in internal solution |
| Software, algorithm | Code used for analyzing MLI firing following photostimulation of single GC | This paper (*Dorgans, 2019a*) | | Python code deposited on GitHub: https://github.com/Dorgans/eLife2018-STP-GC-MLI/blob/master/2017051_GC_photostimulation__MLI_FIRING_ANALYSIS.py |
| Software, algorithm | Code used for analyzing EPSC charge following photostimulation of single GC | This paper (*Dorgans, 2019b*) | | Python code deposited on GitHub: https://github.com/Dorgans/eLife2018-STP-GC-MLI/blob/master/20170711_GC_photostimulation_GC-MLI_CHARGE_ANALYSIS.py |
| Software, algorithm | Code used for PCA transformation and *k*-mean clustering analysis of EPSC charges following photostimulation | This paper (*Dorgans, 2019c*) | | Python code deposited on GitHub: https://github.com/Dorgans/eLife2018-STP-GC-MLI/blob/master/20170718_GC_photostim_MLI_SeqPatch_PCA%2Cclustering.py |

*Continued on next page*

*Continued*

| Reagent type (species) or resource | Designation | Source or reference | Identifiers | Additional information |
|---|---|---|---|---|
| Software, algorithm | Code used for PCA transformation and k-mean clustering analysis of EPSC charges during high frequency stimulations | This paper (*Dorgans, 2019d*) | | Python code deposited on GitHub: https://github.com/Dorgans/eLife2018-STP-GC-MLI/blob/master/20170718_GC_photostim_MLI_SeqPatch_PCA%2Cclustering.py |

## Mice

This study was carried out in strict accordance with the national and international laws for laboratory animal welfare and experimentation and was approved in advance by the Ethics Committee of Strasbourg (CREMEAS; CEEA35; agreement number/reference protocol: APAFIS#4354–20 16030212155187 v3). Mice were bred and housed in a 12 hr light/dark cycle with free access to food and water. Wild type (WT) or Synapsin II knock-out (Syn II KO) mice have CD1 genetic background. Syn II KO mice were first derived from synapsin triple knock-out mice (C57BL/6J genetic background, originating from the Italian Institute of Technology, Genova, Italy) (*Gitler et al., 2004*) bred with CD1 WT mice. Syn II KO hybrid mice were serially bred (10 backcrosses) with CD1 WT mice to obtain Syn II KO mice with CD1 genetic background.

## Slice preparation

Acute cerebellar slices were prepared from CD1 mice or Syn II KO mice (*Rosahl et al., 1995*), aged 20 to 35 days. Mice were anesthetized by isoflurane inhalation and decapitated. The cerebellum was extracted in ice-cold (~1°C) artificial cerebrospinal fluid (ACSF) bubbled with carbogen (95% $O_2$, 5% $CO_2$) containing (in mM): 120 NaCl, 3 KCl, 26 NaHCO3, 1.25 $NaH_2PO_4$, 2.5 $CaCl_2$, 2 $MgCl_2$, 10 D-glucose and 0.05 mM minocyclin. Cerebella were sliced (Microm HM650V, Germany) in an ice-cold low-sodium and zero-calcium slicing buffer containing (in mM): 93, 2.5 KCl, 0.5 CaCl2, 10 MgSO4, 1.2 NaH2PO4, 30 NaHCO3, 20 HEPES, 3 Na-Pyruvate, 2 Thiourea, 5 Na-ascorbate, 25 D-glucose and 1 Kynurenic acid. Sagittal or horizontal slices 300 µm thick were immediately transferred for recovery in a bubbled ACSF for 30 min at 34°C and maintained at room temperature (~25°C) in bubbled ACSF before use.

## Electrophysiology

After at least 1 hour of recovery at room temperature (~25°C), slices were transferred in a recording chamber continuously perfused with 32 ~ 34°C bubbled ACSF. In order to block all forms of long-term synaptic plasticity and trans-synaptic signaling, blockers of GABA$_A$-receptors (100 µM picrotoxin), GABA$_B$-receptors 10 µm (3-[[[(3,4-Dichlorophenyl)- methyl]amino]propyl(diethoxymethyl)phosphinic acid), NMDA-receptor (100 µM D-AP5; D-(-)−2-Amino-5-phosphonopentanoic acid), endocannabinoïd CB1 receptors (1 µM AM251 1-(2,4-Dichlorophenyl)−5-(4-iodophenyl)−4-methyl-N-(piperidin-1-yl)−1H-pyrazole-3-carboxamide) and mGluR1 receptor (2 µM JNJ16259685 (3,4-Dihydro-2H-pyrano[2,3-b]quinolin-7-yl)-(cis-4-methoxycyclohexyl)-methanone) were added in ACSF.

MLI were patch-clamped in lobules IV to VI in the cerebellar vermis using a two-photon microscope setup (Multiphoton Imaging System, Scientifica UK) with 10 MΩ resistance glass electrodes containing a cesium-based intra-cellular medium (140 mM CsCH3SO3, 10 mM Phosphocreatine, 10 mM HEPES, 10 mM BAPTA, 4 mM Na-ATP and 0.3 mM Na-GTP) supplemented with 50 µM Atto-594 fluorescent dye (Sigma-Aldrich, Germany). In all experiments, cells were voltage-clamped at −70 mV in whole-cell configuration (Multiclamp 700B, Molecular Devices). Data were acquired using the WinWCP freeware (John Dempster, Strathclyde Institute of Pharmacy and Biomedical Sciences, University of Strathclyde, UK). Electrical stimulations were realized with a ~ 10 MΩ resistance monopolar electrode also filled with ATTO-594 for a precise adjustment of the distance between the stimulation pipette and isolated dendritic processes of MLIs. Electric pulses were adjusted at any GC-MLI contact and evoked with a stimulator (IsoStim01-D, NPI Germany). Variations of input resistance

were not corrected but monitored. The recordings were not taken into account if serie resistance has changed by more than 20%.

Minimal stimulation was used to monitor STP at unitary GC-MLI synapses in sagittal slices (*Figure 1—figure supplement 1A*). To do so, we followed previously established procedures (*Malagon et al., 2016*). For each synapse, the intensity of electrical stimulation was maintained in an intensity window that avoided both stimulation failures and multiple-synapse stimulation (*Figure 1—figure supplement 1*). Several parameters were systematically checked to choose the optimal value of the stimulating current ensuring the recruitment of a single synapse. First, the stimulation must induce a high success rate (>0.4) and stable EPSC integral at the first stimuli in the train. The lowest current value reaching these criteria was chosen for minimal stimulation. Stimulating currents were rejected if they generated (i) systematic failures at the first stimulus, (ii) an increase in failure rate at the second stimulus, (iii) no decrease in the failure rate at the second stimulus for synapses with a high failure rate (~0.3 to 0.6) at the first stimulus, (iv) a strong (>5 fold) increase in EPSC amplitude at the first stimulus in one or more responses for the same stimulating current, and/or (v) aberrant values of the paired-pulse ratios (>4) during 10 successive stimuli. In almost 80% of experiments, the current values of minimal stimulations were inferior to 50 µA (*Figure 1—figure supplement 1C*). Our results showed that current spread in cerebellar slices follows a biexponential decay function with two current-distance constant ($\tau_1$ = 12.051 µm for ± 4.06 µm that accounts for 93.3% of current drop and $\tau_2$ = 8.8*$10^{-3}$ µm ± 4.2*$10^{-3}$ µm that accounts for the last 6.7% of the drop). Given that intersynaptic distance on MLI dendrites is estimated to 10 µm (*Soler-Llavina and Sabatini, 2006*), nearest synaptic contacts of the recorded synapses received a current 52.9% lower that this latter (*Figure 1—figure supplement 1D*). To summarize, it is likely that current spread properties ensure the recruitment of single synapses when using stimulation intensities just above the threshold.

We performed glutamate-uncaging assays onto horizontal slices by using MOSAiC patterned illumination system (Andor Technologies). MLI were recorded in ACSF containing 100 µM RuBiGlutamate (*Valera et al., 2016*). In order to find connected pairs of GC-MLI, we first used full-field arrays composed of very small photostimulation areas (15 ~ 25 µm diameter) and patches of GC were sequentially illuminated with blue light (460 nm). We took advantage on horizontal slice configuration to stimulate GCs localized at distant locations from the recorded MLI. Considering the weak probability of connection between GC and MLI, synaptic activities evoked by photostimulation of small cluster of GCs localized far away of the recorded MLI are likely to originate from unitary GC-MLI synapses.

## Post-hoc 3D reconstructions

After the experiments, two-photon Z-stacks (1 µm resolution) were done to reconstruct the recorded MLIs in sagittal configuration using the *simple neurite tracer* plugin (*Longair et al., 2011*) from ImageJ freeware (National Institute of Health, USA). Basket cells were identified by the basket-like features observed in the Purkinje cell layer (*Palay and Chan-Palay, 1974*).

## Electron microscopy

CD1 mice aged 20 days (WT and Syn II KO mice) were deeply anesthetized by intra-peritoneal injection of Ketamine (2 ml/kg) and Xylazine (0.5 ml/kg) and intracardiac perfusion was performed with 2.5% glutaraldehyde in phosphate buffer (0.1 M, pH 7.4). For immunogold labeling, the fixative solution was replaced by 0.1% glutaraldehyde and 4% paraformaldehyde in phosphate buffer. Transversal cerebellar vibratome sections (75 µm thick) were cut and processed either for ultrastructural analysis or for pre-embedding immunogold labeling. After three washes in phosphate buffer, sections were post-fixed in phosphate buffer with 1% OsO4 for 1 hr. Slices were dehydrated in a graded alcohol series (ethanol 25%, 50%, 70%, 95% 100%; 10 min per bath) except for ethanol 100% (3 × 10 min) followed by an incubation in propylene oxide for 3 × 10 min. Then slices were embedded in Araldite M (wash in propylene oxide at 1:1 for 1 hr followed by Araldite M for 2 × 2 hr at room temperature; polymerization at 60°C for 3 days). Ultrathin sections were finally contrasted with uranyl acetate.

## Pre-embedding immunogold labelling

Sections were permeabilised with 0.2% saponin in phosphate buffer saline (PBS) for 1 hour, rinsed in PBS and blocked in a blocking solution: 2% bovine serum albumin in PBS (PBS-BSA). The sections were incubated overnight with anti-Syn I (1/250) or anti-Syn II (1/100) antibodies (polyclonal, SynapticSystems) in 0.1% BSA in PBS. After washing in PBS-BSA, the sections were incubated in Ultra small nanogold F(ab') fragments of goat anti-rabbit or goat anti-mouse immunoglobulin G (IgG) (H and L chains; Aurion) diluted 1/100 in PBS-BSA. After several rinses in PBS-BSA and in phosphate buffer (PB), sections were postfixed in glutaraldehyde 2% in PB before washing in PB and distilled water. Gold particles were then silver enhanced using the R-Gent SE-EM kit (Aurion) before being washed in distilled water and PB. Finally, the sections were post-fixed in 0.5% OsO4 in PB for 10 min before classical processing for Araldite embedding (Sigma, St. Louis, MO) and ultramicrotomy. The ultrathin sections were counterstained with uranyl acetate and observed with a Hitachi 7500 transmission electron microscope (Hitachi High Technologies Corporation, Tokyo, Japan) equipped with an AMT Hamamatsu digital camera (Hamamatsu Photonics, Hamamatsu City, Japan). In control sections processed without anti-Syn I or anti-Syn II primary antibodies or gold-labeled secondary antibodies, no gold particles were observed.

## Analysis of electron micrographs

PF-MLI and PF-PC synapses are glutamatergic synapses that can be recognized by the presence of an obvious asymmetry with a large postsynaptic density (*Korogod et al., 2015*). GCs contact MLIs on their dendritic shaft and PCs on their dendritic spines. Since PC dendritic spines are devoid of mitochondrion (*Palay and Chan-Palay, 1976*), GC-MLI synapses stand out from the large majority of asymmetrical synapses in the cerebellar cortex by the presence on mitochondrion within the postsynaptic compartment. Also, we only took in account synapses located to the upper part of the molecular layer to avoid GC-Golgi cell synapses. Morphometric analyses were performed using ImageJ freeware (National Institute of Health). We binned the number of vesicles (with 50 nm distance bins) starting from the active zone cytomatrix as a reference point (0 nm). Synaptic vesicles within 50 nm of the active zone were considered as docked vesicles (*Schikorski and Stevens, 2001*).

## Immunohistochemistry

CD1 WT mice aged 20 to 25 days were deeply anaesthetized by intra-peritoneal injection of Ketamine (2 ml/kg) and Xylazine (0.5 ml/kg) and perfused with PBS containing 4% paraformaldehyde (PFA). After a 3 hr post-fixation, cerebella were sliced in sagittal configuration (50 μm thickness). Slices were washed in PBS (3 × 10 min). Membranes were permeabilized by 0,1% TritonX100 and non-specific antigens were blocked by 10% bovine serum albumin (BSA) and 1% goat serum albumin (GSA) during 6 hr. Synapses were stained using the same solution supplemented with anti-VGluT1 guinea pig polyclonal antibodies diluted at 1/600 (Synaptic Systems, Germany), polyclonal rabbit anti-Syn Ia/SynIb (Synaptic Systems, Germany) diluted at 1/500 and monoclonal anti-Syn IIa/Syn IIb antibodies. We used two different monoclonal anti-Syn IIa/Syn IIb antibodies both diluted at 1/500: clone 27E3 (Synaptic Systems) targeted again an epitope localized on domains C and clone 19.4 (Millipore) targeted again an epitope localized on domain A-B. Secondary antibodies (Abcam) were applied during 3 hr in a solution containing 10% BSA. Slices were mounted and visualized under confocal microscope (Leica SP5, II).

## Data analysis

Analysis were performed with home-made python routines (WinPython 3.3.5, Python Software Fundation) based on custom scripts. All statistical analyses were performed using SciPy plugin (https://scipy.org/) (*Dorgans, 2019a*; *Dorgans, 2019b*; *Dorgans, 2019c*; *Dorgans, 2019d*; see key resource table; copies archived at https://github.com/elifesciences-publications/eLife2018-STP-GC-MLI). Error bars represent ± SEMs of data distribution. Student's t-test was used in the case of a normal distribution of data, Mann-Whitney Rank Sum Test (MWRST) was used in other cases. One way ANOVA with *post hoc* Tukey tests were used for multiple comparisons. The levels of significance are indicated as ns (not significant) when $p > 0.05$, * when $p < 0.05$, ** when $p < 0.01$ and *** when $p < 0.001$.

## Principal component analysis

PCA is a linear transformation algorithm that examines the main sources of variability inside a dataset composed of multiple observations in order to classify the dataset. PCA analyses covariance between the *n* variables of a dataset and transforms an original dataset in *eigenvalues* around a small number of dimensions representing the principal components. The first two Principal Components (PC1 and PC2) which explain the highest source of variance from the original dataset are represented in a scatter plot. We used PCA in order to classify STP in our datasets and extract the most relevant inter-individual differences. PCA were computed using the python-based *sklearn* plugin. Input variables were normalized and centered using Vector Space Model (VSM) that linearly scales the observations between 0 and 1 (*Salton et al., 1975*). While STP data from WT GC-MLI terminals was used for PCA computation, Syn II KO observations did not take part in the eigenvalue calculation. In order to compare STP heterogeneity between the two populations of synapses, Syn II KO observations were processed as additional values and overlaid to WT cloud of points.

## Data processing

For STP analyses using minimal stimulation protocols, data was collected by estimating EPSC charges at any stimulus number from 7 (or more) consecutive trains at 100 Hz elicited every minutes. Failures were arbitrarily detected as signals below a threshold of 3 x $\sigma_{noise}$, where $\sigma_{noise}$ is the standard deviation of the amplitude of the noise calculated on a 300 ms fixed temporal window preceding the stimulation. PCA transformations (*Figures 2*, *5* and *7*) were performed on the median charge value of each EPSC from the 100 Hz train pulse for each synapse (n = 96). The charge of EPSCs evoked at unitary GC-MLI synapses by photostimulation was measured in a minimal number of 7 successive recordings. To calculate the average charge, values were binned (bin width = 5 ms) from the stimulation onset to 100 ms post-stimulus for each sweep (n = 1080) and PCA transformation was applied using the charge value for unitary dataset (*Figures 6* and *7* n = 89). The delay of MLI peak frequency was estimated from the stimulation onset.

In rare cases, photostimulating GCs could evoke neurotransmitter release at more than one GC-MLI synaptic contact. To optimize GC-MLI STP dataset, *post-hoc* monitoring of EPSCs evoked by GC activation was systematically performed using SpAcAn (Spontaneous Activity Analysis), a collection of IGOR pro-functions (WaveMetrics) (https://www.wavemetrics.com/project/SpAcAn). When EPSCs displayed important kinetic variability, the recordings (both loose-patch and whole-cell recordings) were systematically discarded.

## Acknowledgements

This work was supported by the Centre National pour la Recherche Scientifique, the Université de Strasbourg, the Agence Nationale pour la Recherche Grant (ANR-2015CeMod) and by the Fondation pour la Recherche Médicale to PI (# DEQ20140329514). KD was funded by a fellowship from the Ministère de la Recherche. We thank Dr. Sophie Reibel-Foisset and the staff of the animal facility (Chronobiotron, UMS 3415 CNRS and Strasbourg University) for technical assistance. We thank Pr. Fabio Benfenati (Italian Institute of Technology, University of Genova, Genova, Italy) for the gift of synapsin triple knock-out mice. We thank Dr. Frank Pfrieger for critical reading of the manuscript.

## Additional information

### Funding

| Funder | Grant reference number | Author |
| --- | --- | --- |
| Agence Nationale de la Recherche | ANR-2015CeMod | Philippe Isope |
| Fondation pour la Recherche Médicale | DEQ20140329514 | Philippe Isope |
| Ministère de l'Education Nationale, de l'Enseignement Superieur et de la Recherche | | Kevin Dorgans |

The funders had no role in study design, data collection and interpretation, or the decision to submit the work for publication.

## Author contributions

Kevin Dorgans, Conceptualization, Resources, Data curation, Software, Formal analysis, Validation, Investigation, Visualization, Methodology, Writing—original draft, Writing—review and editing; Valérie Demais, Resources, Investigation, Methodology; Yannick Bailly, Methodology, Writing—review and editing, Conceptualization and supervision of electron microscopy experiments; Bernard Poulain, Conceptualization, Methodology, Writing—review and editing; Philippe Isope, Conceptualization, Resources, Software, Formal analysis, Supervision, Funding acquisition, Validation, Visualization, Methodology, Writing—original draft, Project administration, Writing—review and editing; Frédéric Doussau, Conceptualization, Formal analysis, Supervision, Validation, Investigation, Visualization, Methodology, Writing—original draft, Project administration, Writing—review and editing

## Author ORCIDs

Kevin Dorgans http://orcid.org/0000-0003-1724-6384
Bernard Poulain http://orcid.org/0000-0002-2601-5310
Philippe Isope http://orcid.org/0000-0002-0630-5935
Frédéric Doussau https://orcid.org/0000-0002-3769-1402

## Ethics

Animal experimentation: This study was carried out in strict accordance with the national and international laws for laboratory animal welfare and experimentation and was approved in advance by the Ethics Committee of Strasbourg (CREMEAS; CEEA35; agreement number/reference protocol: APAFIS#4354-20 16030212155187 v3).

## Decision letter and Author response

Decision letter https://doi.org/10.7554/eLife.41586.021
Author response https://doi.org/10.7554/eLife.41586.022

# Additional files

## Supplementary files

• Transparent reporting form
DOI: https://doi.org/10.7554/eLife.41586.018

## Data availability

All data generated or analysed during this study are included in the manuscript and supporting files. Python scripts are available at https://github.com/Dorgans/eLife2018-STP-GC-MLI (copies archived at https://github.com/elifesciences-publications/eLife2018-STP-GC-MLI).

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
