## [Decision Letter]

Thank you for sending your article entitled "Short-term plasticity at cerebellar granule cell to molecular layer interneuron synapses expands information processing" for peer review at *eLife*. Your article is being evaluated by three peer reviewers, and the evaluation is being overseen by a Reviewing Editor and Gary Westbrook as the Senior Editor.

A key question arose in the discussion with the editors and reviewers regarding methodology, which we would like to ask you about before we form a decision. Specifically, it was not clear how you determined that action potential failures did not occur during minimal stimulation. The reviewers noted that the 70 µA intensity seemed to give lower proportions of failures than the 80 µA intensity (in Figure 1—figure supplement 1) and pointed out that if stimulation failures indeed occur, the plasticity profiles could be erroneous. We would like to know whether you think that this concern is easily addressed, and if so, how. That point, as articulated by reviewer 1, as well as the other points raised in the reviews, are included below. It is not necessary for you to complete a revision yet, but please consider the reviews and respond within the next two weeks letting us know how you might address the points, including new experiments and timetable for the completion of the additional work as necessary. We will share your responses with the reviewers and then issue a binding recommendation.

*Reviewer #1:*

This work by Dorgans et al. examines short-term plasticity (STP) at single parallel fiber-molecular layer interneuron synapses. They find that STP is highly variable across individual synapses and that responses fall into four types groups using principle component analysis. They then show that expression of synapsin II is also variable across synapses and suggest this may account for at least part of the variability in STP. In syn II KO mice, release probability from PF-MLI synapses is reduced generally, and STP responses fall into only two rather than 4 groups, indicating reduced variability of STP responses. They go on to show that differences in STP at PF-MLI synapses results in correlated differences in the firing response of MLIs to synaptic stimulation. The authors argue that diversity of STP at these synapses increases the coding range of MLIs. The data are generally of high quality and the results are of broad interest. However, the manuscript lacks detail on experimental design and analysis in many instances. In several cases it is difficult to assess the conclusions without further detail.

Major comments:

1) The authors use minimal electrical stimulation to measure STP from single synapses. However, the profile of STP can be highly affected by the efficiency of stimulation. Specifically, failure to reliably elicit an action potential on the first stimulus can result in apparent synaptic failures and alter the apparent STP. Very little detail is given on the stimulus protocol in the Materials and methods, though it does state they avoided stimulation failures, how was this avoided? The Materials and methods cite Figure 1—figure supplement 1 where some detail is provided. From this figure, it looks like 70 and 80 µA stimulation activated a single synapses, but there are far more failures at 70 µA (is this true?). If the same synapse is reliably activated in each case the synaptic failure rate should be the same, suggesting the increased failure rate at 70 µA is due to failure to reliably evoke an AP in the axon. The legend indicates 70 µA was chosen as the minimal stimulation in this case, making me concerned that many recordings may not be reliably stimulating action potentials in the axon on the first stimulus. Was the lowest stimulus intensity to evoke responses always used? And did this intensity have more failures than slightly higher intensities as is shown in Figure 1—figure supplement 1? Malagon et al., 2016 (cited in the Materials and methods) appear to have used the plateau of failure rate to determine the minimal stimulus intensity, but (at least from Figure 1—figure supplement 1) that does not appear to be the case here.

2) Looking at the traces, it appears the synapses vary greatly in the degree of asynchronous release, was this measured or considered as a variable? I would guess that differences in asynchronous release could have a profound impact on STP and the pattern of firing in the post-synaptic MLI. Did the amount of asynchronous release correlate with any of the response groups?

3) More detail should be provided on the principal component analysis. It's not clear to me what a positive or negative PC1 or PC2 value means.

4) Regarding Figure 4. Using immunostaining it is not possible to distinguish PF-MLI synapses from PF-PC synapses, and in fact the majority of synapses are likely PF-PC synapses, this should be acknowledged in the text.

Is it possible to conclusively distinguish PF-MLI synapses form PF-PC synapses in EM? If so, how did you determine this? These details are missing. It is important to know that the synapses being analyzed are in fact PF-MLI synapses.

Have you (or others) tested the reliability of the syn II antibody? Is it possible you see less syn II labeling because the antibody is not as good as the syn I antibody?

5) Figures and text frequently report EPSC charge, how was this calculated?

6) In Figure 1C, does this show the quantification of responses from B? If so, why do responses increase on the last stimulus when this does not appear to be the case in the traces in (B). The green trace in particular, EPSC10 is the same amplitude as EPSC1, but this is hard to believe looking at the green trace in (B). This again brings up the question of how EPSC charge was calculated.

7) Figure 7 C, D. Under loose-patch, data are graphed as "firing frequency acceleration,% ", this is somewhat confusing, is there a reason for not simply graphing the firing rate? Graphs in (D) are "Delay to frequency peak" and "Delay to firing peak", how are these different? Some descriptions of these would help.

8) Second paragraph of subsection “Diversity of STP profile at GC-MLI connections extends the coding range of MLI”, is this shown in Figure 7D? Again, what does PC1 represent here? I don't see that either graph in 7D graphs "peak frequency".

*Reviewer #2:*

Dorgans et al. investigate heterogeneity in synaptic properties of cerebellar granule cells (GC) onto molecular layer interneurons (MLI). The GC-MLI synapse shapes cerebellar cortical computations by recruiting stellate and basket interneurons, thus engaging feedforward inhibition onto Purkinje neurons. Through local stimulation of individual release sites, the authors find that the GC-MLI connection is heterogeneous in terms of its short-term plastic properties. They speculate that this heterogeneity reflects variable initial release probabilities. Unexpectedly, they find that the heterogeneity cannot be explained by target-dependence (stellate cells vs. basket cells), which is known to exist in this circuit, but that it is influenced by presynaptic expression of the synaptic vesicle-associated protein synapsin II: in synapsin II knockout animals, the heterogeneity in short-term synaptic plasticity profiles is reduced. Finally, they show that the diversity in synaptic profiles has functional implications for the differential recruitment of MLI, and could therefore be a mechanism to expand coding of information in the cerebellar cortex.

The study is interesting. A few issues should be addressed:

Experiments are done with very small electrodes (10 MΩ), from cells with very thin dendrites, thus raising concerns about voltage control. There is no mention of series resistance compensation. I wonder to what extent the different synaptic profiles are influenced/shaped by series resistance errors. Is there a relationship between distance of synaptic inputs from cell body and type of synaptic profile?

Synapsins are known to regulate the size of vesicle pools (e.g. Gitler et al., 2008), in addition to release probability. Is paired-pulse plasticity affected in synapsin knockouts? Is the size of the RRP affected? is there an effect on vesicle clustering in EM?

The lack of target cell specificity in the heterogeneity of GC connections on MLI directly contradicts previous findings, which is acknowledged by the authors. One explanation they offer is that in previous studies, strong connections in compound responses (elicited by stimulation of GC clusters or by parallel fiber stimulation) could mask weak responses. The authors can and should directly address this important issue using their GC-specific photostimulation assay. Are there target-dependent differences in photostimulation-evoked synaptic profiles? (they might already have the data to answer this question).

Subsection “Syn II is heterogeneously expressed across GC-MLI presynaptic terminals”: the authors correlate vglut1 signal with synapsin I signal (presumably in the molecular layer, although that is not mentioned) and report R= 0.584. They conclude that "synapsin I is present in ALL GC terminals". I think this conclusion is not justified given the low R-value and should be rephrased.

*Reviewer #3:*

In their study Dorgans et al. use PCA to classify four classes of Cerebellar Granule Cell – Molecular Interneuron Cell synapses based on the short-term dynamics of the MIC responses to repeated stimulation of the GC. They show that if they knockout synapsin II from the GC axon terminals that the dynamics and time course of the synaptic responses are altered and slowed. Their demonstration of the functional significance of this 'heterogeneity' in synaptic responsiveness involved characterisation of the spiking output of the MICs and indeed they show that cells that receive strong initial synaptic drive spike early while those that receive a slower more gradual ramp in synaptic drive spike later. Overall the experiments are carried out to a very high standard and I have no doubts that the results are robust and their interpretation correct.

[Editors' note: further revisions were requested prior to acceptance, as described below.]

Thank you for submitting your article "Short-term plasticity at cerebellar granule cell to molecular layer interneuron synapses expands information processing" for consideration by *eLife*. Your article has been reviewed by two peer reviewers, and the evaluation has been overseen by a Reviewing Editor and Gary Westbrook as the Senior Editor. The reviewers have opted to remain anonymous.

The reviewers have discussed the reviews with one another and the Reviewing Editor has drafted this decision to help you prepare a revised submission.

Summary:

This study examines short-term plasticity (STP) at single parallel fiber-molecular layer interneuron synapses and demonstrates that STP varies across individual synapses, with responses falling into four categories as defined by principal components analysis. Additionally, expression of synapsin II also differs across synapses and may account for at least part of the variability in STP, as demonstrated by tests of synapsin-II knock-out mice. Additionally, differences in STP parallel fiber to interneuron synapses correlates with differences in the firing responses of interneurons to synaptic stimulation.

Essential revisions:

Both reviewers agreed that the revisions have substantially improved the manuscript. One concern remains regarding the matter of stimulation intensity, namely whether each stimulus reliably evoked action potentials in parallel fibers, and it was suggested that it would be appropriate to show in a supplement a plot of failure (or success) rate versus stimulus intensity. In the consultation, it was acknowledged that even if failures were evident, it would not compromise the results and interpretations in a way that could not be addressed by a straightforward discussion and/or acknowledgment of possible sources error. Our consensus was that such a graph is needed simply for complete disclosure of experimental conditions. In the reviewer's words: "What I would really like to ask them to do, is (1) plot failure rate vs stimulus intensity for every cell and exclude cells in which the failure rate has not reached a plateau at the stimulus intensity used, (2) provide a better example in supplement figure 1B including the failure rate plot and (3) add a statement that the stimulus intensity used was the minimum intensity that produced a failure rate of at least 90% (or whatever the real value is) of the plateau failure rate."

The reviewer's original expression of this point are included below.

Major comments:

The authors have substantially responded to the reviewers' concerns and improved the manuscript, including additional experiments, changes to figures, additional supplementary figures, and major changes in the text. I am satisfied with the use of an additional syn II antibody and other responses to reviewer comments. I am reasonably confident that responses result from stimulation of a single synapse and not multiple synapses. The authors use an impressive array of analysis, experiments, and strict response criteria to establish that EPSCs arise from a single synapse.

However, I am less certain that an AP was reliably evoked in the parallel fiber with each stimulus. As I said in the previous review, in Figure 1—figure supplement 1B there are far more failures at 70 µA stimulus that 80 µA stimulus. This strongly suggests AP were not reliably evoked at this stimulus intensity, and a higher stimulus intensity (80 µA) should be used. The authors need to show a plot of failure (or success) rate versus stimulus intensity (as is shown in figure 2D of Malagon et al., 2016) and indicate the stimulus intensity chosen. Ideally, this plot should show that the failure rate reaches a plateau at the stimulus intensity used for subsequent experiments (though I suspect this is not the case for the data in the figure). The authors do mitigate this concern to a degree by showing that when the first EPSC is excluded the principal component analysis still produces more or less the same groups of responses (Figure 2—figure supplement 3). However, I still think the plot of failure rate vs stimulus needs to be shown in Figure 1—figure supplement 1.

---

## [Author Response]

[Editors' note: the authors’ plan for revisions was approved and the authors made a formal revised submission.]Reviewer #1:This work by Dorgans et al. examines short-term plasticity (STP) at single parallel fiber-molecular layer interneuron synapses. They find that STP is highly variable across individual synapses and that responses fall into four types groups using principle component analysis. They then show that expression of synapsin II is also variable across synapses and suggest this may account for at least part of the variability in STP. In syn II KO mice, release probability from PF-MLI synapses is reduced generally, and STP responses fall into only two rather than 4 groups, indicating reduced variability of STP responses. They go on to show that differences in STP at PF-MLI synapses results in correlated differences in the firing response of MLIs to synaptic stimulation. The authors argue that diversity of STP at these synapses increases the coding range of MLIs. The data are generally of high quality and the results are of broad interest. However, the manuscript lacks detail on experimental design and analysis in many instances. In several cases it is difficult to assess the conclusions without further detail.Major comments:1) The authors use minimal electrical stimulation to measure STP from single synapses. However, the profile of STP can be highly affected by the efficiency of stimulation. Specifically, failure to reliably elicit an action potential on the first stimulus can result in apparent synaptic failures and alter the apparent STP. Very little detail is given on the stimulus protocol in the Materials and methods, though it does state they avoided stimulation failures, how was this avoided? The Materials and methods cite Figure 1—figure supplement 1 where some detail is provided. From this figure, it looks like 70 and 80 µA stimulation activated a single synapses, but there are far more failures at 70 µA (is this true?). If the same synapse is reliably activated in each case the synaptic failure rate should be the same, suggesting the increased failure rate at 70 µA is due to failure to reliably evoke an AP in the axon. The legend indicates 70 µA was chosen as the minimal stimulation in this case, making me concerned that many recordings may not be reliably stimulating action potentials in the axon on the first stimulus. Was the lowest stimulus intensity to evoke responses always used? And did this intensity have more failures than slightly higher intensities as is shown in Figure 1—figure supplement 1? Malagon et al., 2016 (cited in the Materials and methods) appear to have used the plateau of failure rate to determine the minimal stimulus intensity, but (at least from Figure 1—figure supplement 1) that does not appear to be the case here.

Failure rate

A stable failure rate is indeed an essential criterion for ensuring that minimal stimulation actually recruits a single synapse. Several works performed at unitary GC-BC or GC-SC synapses report a failure rate ranged from 0 to 0.6 at the first stimulus (Ishiyama et al., 2014; Malagon et al., 2016; Miki et al., 2016). As noted by Malagon and coworkers, “stable, high success rate for the first stimulus (Figure 2D, dots), together with a stable EPSC integral (Figure 2D, circles), indicated reliable AP firing for each stimulus in the presynaptic PF” (Malagon et al., 2016). We used these criteria to adjust and choose the intensity of stimulation. We systematically probed the failure rate and the amplitude of EPSCs at the first stimuli and at the second stimuli (to probe the paired-pulse ratio) at several values of current intensity during 10 successive paired-pulse stimulations (real time measurement of EPSC charge was not possible). The lowest value reaching these criteria was chosen for minimal stimulation. In general this value ranged between 30 to 40 µA. Stimulations meeting the following criteria were not considered valid for minimal stimulation:

– stimulations generating systematic failures at the first stimuli (considered as infra-threshold stimuli).

– stimulations generating strong variations of the failure rate at the first and/or the second stimuli during 10 successive stimuli

– stimulations generating strong variations of EPSC amplitudes at the first and second stimuli during 10 successive stimuli (real-time EPSC amplitudes could be measured and visualized by superimposing the 10 traces)

– stimulations generating variable values of the paired-pulse ratio during 10 successive stimuli (again, real time estimation of the paired pulse ration could be performed during the experiment) -stimulation generating aberrant values of the paired pulse ratio

Despite such precautions, we cannot exclude that some of the failures are due to lack of excitability rather than synaptic processes. By using our protocol to select an appropriate intensity of minimal stimulation, most of the failures due to a lack of excitability are likely to occur at the first response. To estimate possible errors in the classification of inputs due to experimental errors at the first stimulus, we checked whether the classification of unitary GC-MLI synapses was influenced by the first response. Accordingly, we compared the classification of inputs by PCA and k-mean clustering analysis during 100 Hz trains by taking into account the first response in the train or not. Results are now presented in a new supplementary figure (Figure 2—figure supplement 3). As shown on this supplementary figure, the classification of inputs was weakly affected when the first responses were not included in the analysis. This suggests that the weight of the first response in the classification of input is weak and those experimental errors due to lack of excitability, if any, weakly affect this classification.

The cutoff distance of stimulation

To stimulate only a single synapse, the field covered by the supra-threshold stimulation current should be restricted to distance inferior to the intersynaptic distances measured in MLI dendrites. It is possible to estimate the distance covered by the supra-threshold electrical stimulation simply by moving the tip of the stimulation pipette away from the recording electrode. The “cutoff distance” of stimuli was measured using currents of increasing intensities elicited at increasing distances from the recorded electrodes. Results have now been added in the Figure 1—figure supplement 1. In our experimental conditions, the current dropped sharply when propagating into the slice. This drop could be fitted by a biexponential decay (Figure 1—figure supplement 1). In MLI, intersynaptic distance in dendrites was estimated to ~10 µm (Soler-Llavina and Sabatini, 2006). As an example, at 10 µm from the stimulating electrode, a current of 25 µA (value used in the majority of minimal stimulation experiments) drops to 16.8 µA. Since we always choose the minimal value of current intensity, it is likely that a 16 µA drop of the current intensity cannot elicit response in neighboring synapses.

Despite all the precautions taken, it could be argued that some stimulation could have systematically recruited 2 or more PFs in a stable manner without the experimenter being able to detect it. If so, this phenomenon is supposed to be associated with strong synaptic weights at the first stimuli in the train and is likely to occur for high current values. Conversely, weak current values can generate unexpected failures of PF excitability. This phenomenon is supposed to be associated with large value of PPR since failure of excitability decrease at the second stimuli. Hence, high and low current values may be associated with specific STP profile. We checked whether the charges of EPSCs at the first stimuli, PPR, PC1 and PC2 where correlated with the current intensity. Our result failed to show such correlations. These results are now displayed in Figure 2—figure supplement 2. The lack of correlation between these parameters and the intensity of current indicated that in most of the experiments, STP profile was shaped by synaptic mechanisms rather than change in PF excitability.

Stability of STP profile during 10 successive100 Hz trains

Because high-frequency stimulation can change the excitability of PFs, it should be ensured that no additional PF is recruited during the train. Given the precaution made to select the lowest stimulation intensity, such recruitment is supposed to occur randomly during the train (that is, not systematically and/or at different stimulus number from one train to another). Consequently, recruitment of additional PFs may strongly affect the profile of STP from one train to another. We checked whether the profile of STP was conserved during 10 successive 100 Hz trains at a given position of the stimulation electrode. The results are displayed in Figure 2—figure supplement 2. It should be noted that PCA transformation was performed blindly on every single trace and not on the mean values of ten successive trains. The systematic narrow clustering of 10 recordings belonging to the same series of stimulations in the cloud of point of PCA transformation (Figure 2—figure supplement 2A) clearly indicated that the profile of STP was very stable from one train to another at a given position of the stimulation electrode.

2) Looking at the traces, it appears the synapses vary greatly in the degree of asynchronous release, was this measured or considered as a variable? I would guess that differences in asynchronous release could have a profound impact on STP and the pattern of firing in the post-synaptic MLI. Did the amount of asynchronous release correlate with any of the response groups?

Asynchronous release is indeed variable from synapse to synapse and generally increases during train of high-frequency stimulations at GC-MLI synapses. Measurements of peak amplitude of EPSC only take in account quanta that are released synchronously and therefore underestimate release if a substantial number of quanta are released asynchronously. As mentioned in several studies (for a review, see Neher, 2015), asynchronous and synchronous events are both taken into account by measuring EPSC charge (current integral) instead of EPSC peak amplitude at each stimulus number. In this study, release events and STP were estimated by measuring EPSC charges. Hence, we are quite confident that STP profile could be correctly compared between all types of synapses exhibiting asynchronous release or not.

3) More detail should be provided on the principal component analysis. It's not clear to me what a positive or negative PC1 or PC2 value means.

In our analysis, input variables were normalized and centered. This linearly scaled the values between -1 and 1 (Salton et al., 1975). PCA computes a linear dimensionality reduction on the original dataset using singular value decomposition ARPACK algorithm and computes eigenvectors from n-dimensional dataset (here, n=10; the charge values for eEPSC1 to eEPSC10) composed of multiple observations (x=116 synapses). The eigenvectors are used to transform original dataset into n’-eigenvalues through covariance reduction and variables are sorted from 1 to n’ (here, PC1 to PC5) where PC1 eigenvalues represent maximal variance into the dataset (and PC2 value, the second maximal variance, PC3 the third, etc.). Each Principal Component is explained by a relative proportion of variance from the n-dimensions of original dataset. Thus, calculating the Principal Components of a n-dimensional dataset is the perfect way to illustrate dissimilarities between x-observations by describing non-linearities between individual observations and neglecting co-varying parameters through a global dataset computation.

4) Regarding Figure 4. Using immunostaining it is not possible to distinguish PF-MLI synapses from PF-PC synapses, and in fact the majority of synapses are likely PF-PC synapses, this should be acknowledged in the text.Is it possible to conclusively distinguish PF-MLI synapses form PF-PC synapses in EM? If so, how did you determine this? These details are missing. It is important to know that the synapses being analyzed are in fact PF-MLI synapses.

We agree that in the molecular layer, the vast majority of VGluT1 positive synapses are PF-PC synapses; this is now acknowledged in the present version of the manuscript with the following sentence “Since the vast majority of GC synapses stained by VGluT1 actually correspond to GC-PC synapses, we could not exclude that Syn II is homogeneously expressed in GC-MLI synapses.” The heterogeneous expression of Syn II in GC-MLI synapses was confirmed by EM experiments Concerning the method to recognize PF-MLIs synapses, we also added the following precisions (Material and Methods section):

“PF-MLI and PF-PC synapses are glutamatergic synapses that can be recognized by the presence of an obvious asymmetry with a large postsynaptic density (Korogod et al., 2015). GCs contact MLIs on their dendritic shaft and PCs on their dendritic spines. Since PC dendritic spines are devoid of mitochondrion (Palay and Chan-Palay, 1976), GC-MLI synapses stand out from the large majority of asymmetrical synapses in the cerebellar cortex by the presence on mitochondrion within the postsynaptic compartment. Also, we only took into account synapses located to the upper part of the molecular layer to avoid GC-Golgi cell synapses.”.

Have you (or others) tested the reliability of the syn II antibody? Is it possible you see less syn II labeling because the antibody is not as good as the syn I antibody?

This is an important point. Indeed, we cannot excluded that anti-Syn II antibody (monoclonal antibody from Synaptic System) is less good that anti-Syn I antibodies.

To confirm the heterogeneous expression of Syn II in GC boutons, we performed a new set of immunohistochemical experiments with a Syn II antibody produced by another company (Millipore instead of Synaptic System). It should be noted that the use of Millipore anti-Syn II antibodies required to perform a heat-induced epitope retrieval protocol (slices were incubated 3 time in a boiling citrate buffer at pH 6 during 5 minutes) because Syn II signals were too weak with a regular protocol of staining. This retrieval protocol increased substantially the background noise for both VGluT1 and Syn II staining. Nevertheless, profile plots performed in the molecular layer of cerebellar sagittal sections using Millipore anti-Syn II confirmed the presence of i) VGluT1(+)/Syn II(+) puncta (presence of Syn II in GC boutons), ii) VGluT1(+)/Syn II(i) puncta (absence of Syn II in GC bouton) and iii) VGluT1(-)/Syn II(+) puncta (presence of Syn II in inhibitory synapses) (Author response image 1). The fact that anti-Syn II antibodies recognizing different epitopes of Syn II domains stained only subset of GC boutons strongly suggests that Syn II is actually expressed heterogeneously in GC presynaptic terminals.

**Author response image 1. respfig1:** Heterogeneous expression of Syn II at GC-MLI synapses. (**A**) Domain structure of mammalian Syn IIa and Syn IIb. The red and blue bar represent the domains used as immunogen to produce Millipore and Synaptic Systems (SS) Anti-Syn II antibodies. (**B**) Representative merged images of Syn II (Alexa-488, green) /VGluT1 (Alexa-55, red) immunostaining using Millipore anti-Syn II antibodies. Yellow puncta denote a presence of Syn II in GC boutons while red puncta denote an absence of Syn II in these synapses. Green puncta correspond to Syn II+ inhibitory synapses devoid of VGluT1. Images were captured in the molecular layer from a cerebellar section. The profile plot (blue line) confirms the presence of GC boutons devoid of SynII (red peaks not associated with a green peak). Calibration bar (left image): 7.5 μm. Anti-Syn II: Millipore #MABN1573, clone 19.4 Purified mouse monoclonal IgG2aκ antibody Immunogen: Purified recombinant rat Syn II Epitope: Domains A and B used at 1/500. Anti-Syn II: Synaptic Systems #106 211, clone 27E3 Purified mouse monoclonal IgG antibody Immunogen: Synthetic peptide corresponding to AA 440 to 458 from rat Syn II Epitope: AA 440 to 458 from rat Syn II used at 1/500

5) Figures and text frequently report EPSC charge, how was this calculated?

EPSC charges were calculated as the signal (current) integral in time-locked windows. The window corresponds to the interstimulus interval and excluded stimulus interval. The offset was previously subtracted as well as the mean charge of the noise (measured of a time-locked window preceding the stimulation).

6) In Figure 1C, does this show the quantification of responses from B? If so, why do responses increase on the last stimulus when this does not appear to be the case in the traces in (B). The green trace in particular, EPSC10 is the same amplitude as EPSC1, but this is hard to believe looking at the green trace in (B). This again brings up the question of how EPSC charge was calculated.

Figure 1C does show the quantification for the traces displayed in 1B. There was obviously a mismatch between these traces and the data. We re-measured amplitudes of EPSCs for the traces in B and found different values. The graph in C has been corrected. We thank the reviewer for this careful inspection of the figure.

7) Figure 7 C, D. Under loose-patch, data are graphed as "firing frequency acceleration,% ", this is somewhat confusing, is there a reason for not simply graphing the firing rate? Graphs in (D) are "Delay to frequency peak" and "Delay to firing peak", how are these different? Some descriptions of these would help.

We recognize that explanations were not clear for Figures 7C-D. In slices, MLI are firing spontaneously at low frequencies. Upon single GC photostimulation, the basal rate of MLI firing is accelerated with a variable delay after the onset of photostimulation. The “delay to frequency peak” corresponds to the time separating the onset of photostimulation and the time the recorded MLI is firing is at its maximum frequency. The y-axis is the same for the 2 panels in Figure 7D. Explanations were implemented in the legend as well as the label of y-axis.

8) Second paragraph of subsection “Diversity of STP profile at GC-MLI connections extends the coding range of MLI”, is this shown in Figure 7D? Again, what does PC1 represent here? I don't see that either graph in 7D graphs "peak frequency".

This was a mistake. The sentence “Our analysis also showed a clear relationship between PC1 and the peak frequency (Pearson coefficient, R= 0.7, p = 0,008, n = 13)” is not related to Figure 1D. The correlation between PC1 and the delay to frequency peak was not shown on this figure. In photostimulation experiment, PC1 values drawn from measurement of EPSC charge in whole cell configuration correspond to the best parameter to express variations in STP profiles.

Reviewer #2:Dorgans et al. investigate heterogeneity in synaptic properties of cerebellar granule cells (GC) onto molecular layer interneurons (MLI). The GC-MLI synapse shapes cerebellar cortical computations by recruiting stellate and basket interneurons, thus engaging feedforward inhibition onto Purkinje neurons. Through local stimulation of individual release sites, the authors find that the GC-MLI connection is heterogeneous in terms of its short-term plastic properties. They speculate that this heterogeneity reflects variable initial release probabilities. Unexpectedly, they find that the heterogeneity cannot be explained by target-dependence (stellate cells vs. basket cells), which is known to exist in this circuit, but that it is influenced by presynaptic expression of the synaptic vesicle-associated protein synapsin II: in synapsin II knockout animals, the heterogeneity in short-term synaptic plasticity profiles is reduced. Finally, they show that the diversity in synaptic profiles has functional implications for the differential recruitment of MLI, and could therefore be a mechanism to expand coding of information in the cerebellar cortex.The study is interesting. A few issues should be addressed:Experiments are done with very small electrodes (10 MΩ), from cells with very thin dendrites, thus raising concerns about voltage control. There is no mention of series resistance compensation. I wonder to what extent the different synaptic profiles are influenced/shaped by series resistance errors. Is there a relationship between distance of synaptic inputs from cell body and type of synaptic profile?

Variations of input resistance were not corrected but monitored. The recordings were not taken into account if series resistance has changed by more than 20%.

Since we systematically measured the position of the stimulation pipette from MLI soma, we were able to establish the relationship between the distance of synaptic inputs from the cell body and STP profile. The corresponding graph was added in Figure 3—figure supplement 1 (panel C). The lack of correlation between the profile of STP (estimated with PCA transformation) and the distance from the soma suggested that STP is not determined by the position of input in the dendritic tree of recorded MLIs. A recent study has shown that the properties of excitatory inputs integration by MLI dendrite are equal for inputs localized at distance superior to 20 µm from the MLI’s soma (Tran-Van-Minh et al., 2016). In our experiments, all inputs but one were localized at distance superior to 20 µm from MLI’s soma.

Synapsins are known to regulate the size of vesicle pools (e.g. Gitler et al., 2008), in addition to release probability. Is paired-pulse plasticity affected in synapsin knockouts? Is the size of the RRP affected? Is there an effect on vesicle clustering in EM?

The strong increase in the failure rate observed at the first stimuli in Syn II KO synapses suggests that the probability of release (pr) is impaired. At many synapses, change in the paired-pulse ratio is indeed a good parameter to probe change in pr. However, the release machinery of GC terminal stands out by its capacity to recruited new sites (Miki et al., 2016) or reluctant synaptic vesicles (Doussau et al., 2017) in a millisecond time scale during paired pulse stimulation at high frequencies meaning that at these synapses, an increase in n (number of sites) underlies part of the large facilitation that occurs during paired pulse facilitation (Miki et al., 2016; Valera et al., 2012). The paired-pulse ratio was not significantly increased in Syn II KO mice (mean/median PPR for WT =

1.8/1.6 ± 0.08, n= 101 and mean/median PPR for Syn II KO mice = 2.6/1.7 Syn II ± 0.08 KO, p = 0.19, note that median are similar). These data have been added in the manuscript. This suggests that in Syn II KO mice, the recruitment of the reluctant pool at the second stimuli is normal and mask effect of low pr of the fully releasable pool on PPR (fully releasable vesicles = vesicles released by a single AP, see Doussau et al., 2017).

In Syn II KO mice, the number of docked synaptic vesicles (SVs) is reduced in comparison to WT synapses (Figure 6). The pool of docked SVs comprised unprimed and primed SVs (RRP). The RRP itself is segregated in heterogeneous population of primed SVs (Doussau et al., 2017; Neher and Brose, 2018). Electron micrographs from brain slices cannot permit to distinguish between the different pools of docked SVs and therefore give only partial information about the ultrastructure-function relationship. Nevertheless the reduction of the number of docked SVs in Syn II KO synapse may underlie, at least in part, the increase in the failure rate.

The lack of target cell specificity in the heterogeneity of GC connections on MLI directly contradicts previous findings, which is acknowledged by the authors. One explanation they offer is that in previous studies, strong connections in compound responses (elicited by stimulation of GC clusters or by parallel fiber stimulation) could mask weak responses. The authors can and should directly address this important issue using their GC-specific photostimulation assay. Are there target-dependent differences in photostimulation-evoked synaptic profiles? (they might already have the data to answer this question).

This is an important issue. We addressed this question in the new version of the manuscript. We were able to identified 2 basket cells and 1 stellate cell (post hoc reconstruction) for which we recorded synaptic responses following photostimulation at 3 different locations (that is, 3 different GCs). These results were now presented in a new supplementary figure (Figure 7—figure supplement 2). The result showed clearly that either BCs or SCs are contacted by excitatory connections belonging to different classes. Photostimulation experiments thus confirmed the behavioral heterogeneity of excitatory synaptic inputs contacting same MLIs. We also performed a new set of experiments using compound stimulations of clusters of GCs or beams of PFs associated with post hoc identification of MLI subtypes (Figure 3—figure supplement 1). Our experiments failed to show a target dependence of STP at GC-MLI synapses.

Subsection “Syn II is heterogeneously expressed across GC-MLI presynaptic terminals”: the authors correlate vglut1 signal with synapsin I signal (presumably in the molecular layer, although that is not mentioned) and report R= 0.584. They conclude that "synapsin I is present in ALL GC terminals". I think this conclusion is not justified given the low R-value and should be rephrased.

In presynaptic terminal, VGluT1 is supposed to be exclusively associated to synaptic vesicles while synapsin are associated to synaptic vesicles or cytosolic depending of their phosphorylation status. Hence, even if both proteins are present in the same terminal they are not supposed to be perfectly localized. This explains the value of the Pearson coefficient for Syn I/ VGluT1 colocalization. We agreed that correlation values with the Pearson coefficient are confusing and we rephrased this part.

References

Neher, E. (2015). Perspective Merits and Limitations of Vesicle Pool Models in View of Heterogeneous Populations of Synaptic Vesicles. Neuron *87*, 1131–1142.

Neher, E., and Brose, N. (2018). Dynamically Primed Synaptic Vesicle States: Key to Understand Synaptic Short-Term Plasticity. Neuron *100*, 1283–1291.

[Editors' note: further revisions were requested prior to acceptance, as described below.]

Major comments:The authors have substantially responded to the reviewers' concerns and improved the manuscript, including additional experiments, changes to figures, additional supplementary figures, and major changes in the text. I am satisfied with the use of an additional syn II antibody and other responses to reviewer comments. I am reasonably confident that responses result from stimulation of a single synapse and not multiple synapses. The authors use an impressive array of analysis, experiments, and strict response criteria to establish that EPSCs arise from a single synapse.However, I am less certain that an AP was reliably evoked in the parallel fiber with each stimulus. As I said in the previous review, in Figure 1—figure supplement 1B there are far more failures at 70 µA stimulus that 80 µA stimulus. This strongly suggests AP were not reliably evoked at this stimulus intensity, and a higher stimulus intensity (80 µA) should be used. The authors need to show a plot of failure (or success) rate versus stimulus intensity (as is shown in Figure 2D of Malagon et al., 2016) and indicate the stimulus intensity chosen. Ideally, this plot should show that the failure rate reaches a plateau at the stimulus intensity used for subsequent experiments (though I suspect this is not the case for the data in the figure). The authors do mitigate this concern to a degree by showing that when the first EPSC is excluded the principal component analysis still produces more or less the same groups of responses (Figure 2—figure supplement 3). However, I still think the plot of failure rate vs stimulus needs to be shown in Figure 1—figure supplement 1.

Minimal stimulation

We carefully reexamined all parameters concerning minimal stimulations (failure rate, stimulus intensity, paired-pulse ratio) for all synapses that were included in our analysis in the last version of the manuscript. In this novel analysis, we increased the stringency of the criteria applied. In this new analysis, we then excluded 19 synapses (new n=96). We excluded synapses for which the failure rate dropped abruptly at current intensities below the intensity chosen for minimal stimulation as they may potentially be a signature of a recruitment of several synaptic contacts. These experiments were initially included because the increase in current intensity was not associated with an increase in the mean EPSC amplitude at the first stimulus. We also excluded synapses for which the failure rate was inferior to 0.4.

Note that the classification obtains by k-mean clustering analysis depends on sample size. Accordingly, k-mean clustering analysis changed the identity of a few synapses. However, no change in the main conclusion was observed: the same 4 classes of synapses were equally distributed in WT mice and unequally distributed in Syn II KO mice (near complete disappearance of cluster C1 in KO mice).

Figures 2, 5 and Figure 2—figure supplement 3 have been modified following modification of the dataset (pie charts, PCA analysis, means values of EPSC …)

Example of Figure 1—figure supplement 1

We agree that the example chosen in the previous version of the manuscript to illustrate minimal stimulation was not the most appropriate one. As suggested by the reviewers, we now show the plot of the success rate versus stimulus intensity, not only for this example (panel C) but also for all synapses that were included in the dataset for PCA and k-mean clustering analysis (panel B).